# In-depth human plasma proteome analysis captures tissue proteins and transfer of protein variants across the placenta

Maria Pernemalm[1,3†], AnnSofi Sandberg[1†], Yafeng Zhu[1,3], Jorrit Boekel[1,3], Davide Tamburro[1], Jochen M Schwenk[2], Albin Björk[1], Marie Wahren-Herlenius[1], Hanna Åmark[1], Claes-Göran Östenson[1], Magnus Westgren[1], Janne Lehtiö[1,3*]

[1]Karolinska Institute, Stockholm, Sweden; [2]Royal Institute of Technology, Stockholm, Sweden; [3]Proteogenomics, Science for Life Laboratory, Sweden

**Abstract** Here, we present a method for in-depth human plasma proteome analysis based on high-resolution isoelectric focusing HiRIEF LC-MS/MS, demonstrating high proteome coverage, reproducibility and the potential for liquid biopsy protein profiling. By integrating genomic sequence information to the MS-based plasma proteome analysis, we enable detection of single amino acid variants and for the first time demonstrate transfer of multiple protein variants between mother and fetus across the placenta. We further show that our method has the ability to detect both low abundance tissue-annotated proteins and phosphorylated proteins in plasma, as well as quantitate differences in plasma proteomes between the mother and the newborn as well as changes related to pregnancy.

DOI: https://doi.org/10.7554/eLife.41608.001

*For correspondence:
janne.lehtio@ki.se

†These authors contributed equally to this work

Competing interests: The authors declare that no competing interests exist.

## Introduction

Several studies have presented draft maps of the human tissue proteome using mass spectrometry (MS)-based methods (*Kim et al., 2014*; *Wilhelm et al., 2014*; *Bekker-Jensen et al., 2017*). Due to major analytical challenges in plasma, extensive MS-based plasma proteome studies have largely focused on meta-analysis of publicly available datasets. Examples include the Plasma Proteome Database (PPD) (*Nanjappa et al., 2014*) and the PeptideAtlas plasma proteome repository (*Schwenk et al., 2017*). The recently curated plasma PeptideAtlas now compiles 178 MS datasets, to describe a total of 3509 proteins in plasma, including important in-depth plasma proteomics data-sets from recent years (*Keshishian et al., 2015*; *Geyer et al., 2016a*; *Geyer et al., 2016b*) illustrating state of the art of proteome-wide MS-based plasma proteomics.

One major challenge in plasma proteomics is the ability to analyze larger sample cohorts while retaining proteome coverage. Plasma datasets are often characterized by a high proportion of one-peptide identifications resulting in poor overlap between analyses. For example, in the Plasma Proteome Database 10 546 proteins have been reported from 509 publications; however, only 3784 of the identified proteins are present in two or more studies (*Nanjappa et al., 2014*).

The main analytical challenge when applying proteomics methods to plasma is the presence of a few extremely high abundant proteins that dominate the protein content. Roughly 55% of the total protein mass in plasma is made up by albumin alone and as few as seven proteins together make up 85% of the total protein mass. This can be compared with estimates from tissue and cellular data where 2300 housekeeping proteins are thought to make up 75% of the protein mass (*Kim et al., 2014*).

In general, high plasma proteome coverage is dependent on extensive fractionation. Consequently, aiming for high number of identified proteins comes at the cost of low sample throughput

**eLife digest** Blood cells travel through the blood vessels in a soupy mixture of proteins called plasma. Most of these proteins are plasma-specific, yet small amounts of proteins can leak into the plasma from other body parts and may provide hints about what is going on elsewhere in the body. This could allow doctors to use plasma samples to assess health or detect disease. But so far developing methods to detect these leaked proteins has proved difficult.

Plasma passing through the placenta can transfer proteins between a pregnant woman and her baby. Learning more about these protein exchanges may help scientists understand how the mother and baby adapt to each other and what triggers child birth. But, so far, they have been hard to study. Using DNA to help trace the origins of proteins found in mother or baby could make it easier.

Now, Pernemalm et al. have used DNA sequencing in combination with protein analysis to identify proteins passed between two pregnant mothers and their babies. Comparing the genetic sequences of each mother and child made it possible to trace the origin of the proteins. For example, if a mother had a version of the protein that matched genes the child inherited from its father, they knew it passed from the baby to the mother. This approach found 24 proteins in plasma from two pregnant mothers that had likely passed through the placenta during pregnancy. Pernemalm et al. also analyzed the plasma of 30 healthy individuals and confirmed that it contained several proteins that had likely leaked from other organs, including the lungs and pancreas.

Monitoring protein transfer between pregnant mother and baby may help scientists identify what triggers normal or premature deliveries. One advantage of the technique developed Pernemalm et al. is that it can analyze plasma proteins from large numbers of people, which could enable larger studies. More refinement of the technique may also allow scientists to identify leaked proteins in the plasma that provide an early warning of cancer or other diseases.

DOI: https://doi.org/10.7554/eLife.41608.002

and vice versa. Hence, when larger cohorts have been analyzed the number of proteins monitored drastically decreases. Examples of high-throughput studies include 342 proteins monitored in 230 samples in a longitudinal study of twins using SWATH-MS (*Liu et al., 2015*) or relative quantification of 146 proteins in 500 subjects using iTRAQ labeling (5% peptide FDR) (*Cole et al., 2013*).

Efforts to increase the throughput by shortening the analysis time of shotgun MS have also been made, for example by quantifying 285 proteins from 10 plasma samples in as little as 3 hr analysis time/sample (*Geyer et al., 2016a*). The same technology was later applied to a longitudinal study with 43 patients followed over 12 months, making in total 319 plasma samples measured in quadruplicates (a total of 1294 samples), detecting on average 437 proteins per individual (*Geyer et al., 2016b*).

In 2015, a study by *Keshishian et al. (2015)* reported in total over 5000 proteins from 16 plasma samples using high pH reversed phase separation in combination with iTRAQ 4-plex labeling, making it the single most comprehensive plasma proteomics study to-date.

However, despite these advances in the field, in-depth analysis of clinically-relevant-sized plasma cohorts remains a challenge.

Here, we present a method to achieve unbiased, reproducible, in-depth plasma proteome analysis. High-resolution isoelectric focusing based pre-fractionation prior MS analysis (HiRIEF LC-MS/MS) has demonstrated capability of in-depth proteome analysis of cell and tissue samples (*Branca et al., 2014*). We demonstrate that further development of HiRIEF facilitate in-depth plasma analysis to allow powerful liquid biopsy proteomics. Encouraged by the results, we performed a plasma proteogenomics analysis (*Nesvizhskii, 2014*) to explore the possibility of detecting single amino acid variants (SAAV) in plasma. Presently, only a limited number of studies have applied proteogenomics to plasma (*Liu et al., 2015*; *Johansson et al., 2013*; *Chen et al., 2012*) and as of yet, no global studies have been performed to integrate genomic sequence information to the protein sequence variants in plasma to detect SAAV or other protein sequence alterations, as here presented as proof-of-principle.

## Results

### Optimizing the hirief method for plasma analysis

In the adaptation of the HiRIEF-LC-MS/MS method to plasma analysis, we analyzed female plasma, depleted of high-abundance plasma proteins, using a label-free MS approach. The optimization of the method included evaluation of the pH range of the peptide isoelectric focusing and amount of peptide sample load onto the strips; leading to in total 16 different HiRIEF conditions being assessed (*Table 1*, *Supplementary file 1*). Additionally, the effect of MS analysis time alone on plasma proteome coverage was evaluated by an MS runtime control composed of 72 LC-MS/MS cycles of a depleted sample using identical settings and total MS time as for the HiRIEF fractions (*Figure 1a*). This was done to make sure that the number of identifications post fractionation was not purely an inflation caused by MS analysis time alone or an effect of spurious and possibly false identifications caused in the search pipeline when searching a large number of MS-injections in parallel.

When optimizing the HiRIEF methodology we identified on average 1505 proteins per condition (min 904, max 1888), with strict 1% FDR cut-off at PSM, peptide and protein level (*Table 1*, *Supplementary file 1*) In comparison, the MS runtime control identified as few as 241 proteins. In the runtime control, less than 10 proteins composed 50% of the total protein intensity (*Figure 1— figure supplement 1a*), hence indicating repetitive detection of high-abundance proteins in absence of further fractionation despite having depleted the 14 most abundant proteins.

From the HiRIEF pH range evaluation of broad (pH 3–10), narrow (pH 3.7–4.9) and ultra-narrow ranges (pH 3.7–4.05, 4.0–4.25, 4.2–4.45), we concluded that the pH range did not impact largely on protein identification numbers, which is in line with previously reported studies using HiRIEF on cells and tissue (*Table 1*) (*Zhu et al., 2018*). However, almost twice as many peptides were identified in the broad range compared to the ultra-narrow ranges, implying its usefulness for peptide-based quantitative analyses. As expected, the identification overlap across the different pH intervals was

**Table 1.** Summary of the evaluated conditions in the optimization in terms of number of protein identifications (PSM-, peptide- and protein level FDR 1%).

| Experiment | HiRIEF pH range | Depleted plasma (mg) | # PSM | # Peptides | # Proteins |
|---|---|---|---|---|---|
| MS runtime control | NA | 0.072 | 148304 | 2588 | 241 |
| 1 | 3.0–10.0 | 1.0 | 123262 | 15424 | 1626 |
| 2 | | 0.6 | 118983 | 16372 | 1791 |
| 3 | 3.7–4.9 | 1.0 | 72287 | 8566 | 1517 |
| 4 | 3.7–4.05 | 0.2 | 25033 | 3230 | 1068 |
| 5 | | 0.6 | 25472 | 3871 | 1394 |
| 6 | | 1.0 | 39351 | 5421 | 1774 |
| 7 | 4.0–4.25 | 0.2 | 29224 | 2916 | 904 |
| 8 | | 0.6 | 47470 | 4629 | 1534 |
| 9 | | 1.0 | 64970 | 6012 | 1888 |
| 10 | | 1.0 | 46230 | 4695 | 1360 |
| 11 | | 2.0 | 53323 | 5324 | 1659 |
| 12 | | 4.0 | 71727 | 6339 | 1812 |
| 13 | | 6.0 | 70376 | 6149 | 1608 |
| 14 | 4.2–4.45 | 0.2 | 30067 | 2992 | 913 |
| 15 | | 0.6 | 41619 | 4470 | 1422 |
| 16 | | 1.0 | 63329 | 6042 | 1803 |
| A549 cell line* | 3.0–10.0 | — | 367570 | 141071 | 9816 |
| A549 cell line* | 3.7–4.9 | — | 314305 | 93329 | 9679 |

*Cell line data from *Zhu et al. (2018)*.

DOI: https://doi.org/10.7554/eLife.41608.011

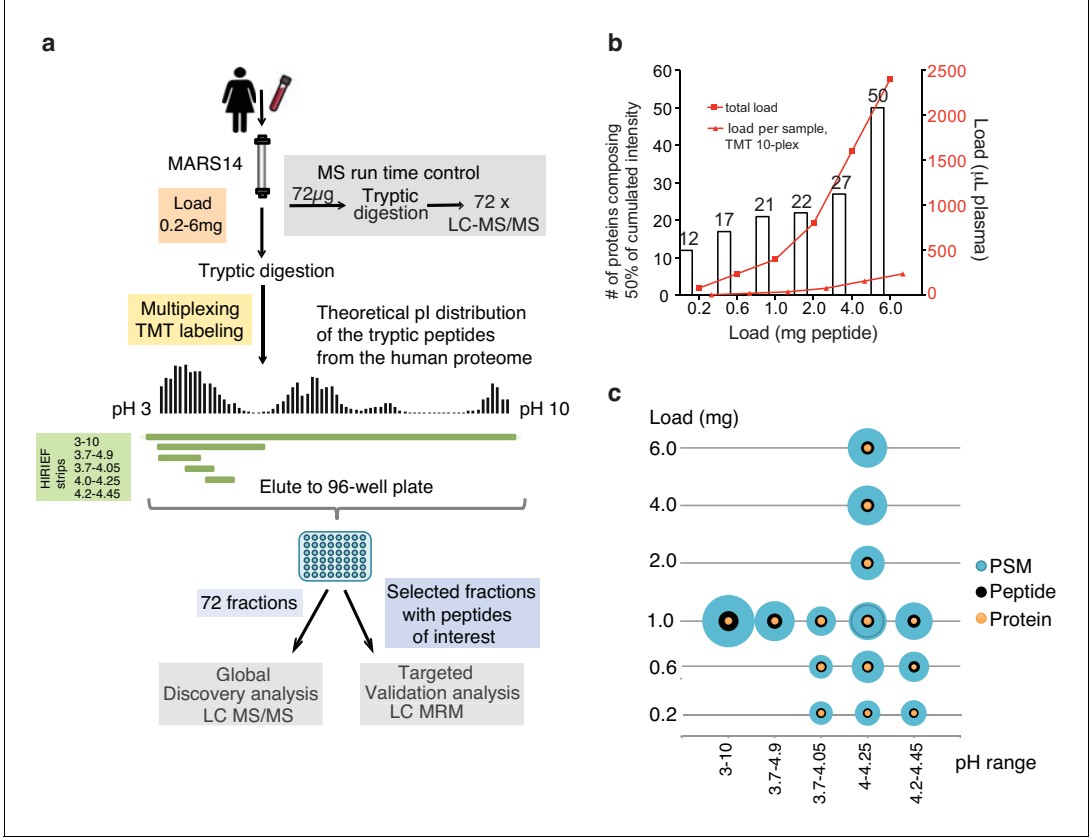

**Figure 1.** Plasma HiRIEF method optimization and performance assessment. (**a**) An overview of the method optimization workflow. Prostate specific antigen was spiked into female plasma at 4 ng/mL. The plasma was depleted and digested and total peptide sample loads between 0.2 and 6 mg were evaluated on five different HiRIEF pH ranges. See *Supplementary file 1* for a summary of the evaluated conditions. For multiplexing and reproducibility calculations tandem mass tags (TMT)-labeling was applied. Fractionated samples were evaluated based on the number of protein identifications using LC-MS/MS and analytical depth using LC-MRM analysis of fractions containing PSA peptide. (**b**) The effect of sample load on analytical depth. The bar chart shows the number (#) of identified proteins making up 50% of total sample intensity (MS1 peak area) as a function of sample loaded onto the HiRIEF pH strips. The corresponding amount of crude plasma used in each condition is shown as read lines (**c**) Combined load and pH range evaluation. Bubble plots showing the effect of the sample load and the HiRIEF pH range on the number of peptide spectrum matches (PSMs), peptide and protein identifications, respectively. The size of the bubbles is proportional to the number of identifications in each experiment.
DOI: https://doi.org/10.7554/eLife.41608.003

The following figure supplements are available for figure 1:

**Figure supplement 1.** Comparison of overlap and performance between different pI ranges.
DOI: https://doi.org/10.7554/eLife.41608.004

**Figure supplement 2.** Venn diagram showing protein (gene centric) overlap between plasma HiRIEF data, human plasma PeptideAtlas build 2017 and plasma proteins currently targeted with affinity proteomics assay.
DOI: https://doi.org/10.7554/eLife.41608.005

**Figure supplement 3.** Proportional distribution of extracellular, membrane, cytoplasmic and nuclear proteins detected in plasma using HiRIEF (all pH ranges, alone or summed), no fractionation (MS runtime control) or the compiled data sets in the plasma PeptideAtlas build 2017.
DOI: https://doi.org/10.7554/eLife.41608.006

**Figure supplement 4.** Use of theoretical peptide pI in global and targeted plasma HiRIEF.
DOI: https://doi.org/10.7554/eLife.41608.007

**Figure supplement 5.** Peptide spread across HiRIEF fractions.
DOI: https://doi.org/10.7554/eLife.41608.008

**Figure supplement 6.** Reproducibility assessment, 'Healthy donors cohort'.
DOI: https://doi.org/10.7554/eLife.41608.009

**Figure supplement 7.** Reproducibility assessment, 'Female longitudinal cohort'.
DOI: https://doi.org/10.7554/eLife.41608.010

larger on protein level than on peptide sequence level (*Figure 1—figure supplement 1b*). The overlapping proteins between different pH intervals were predominantly annotated as high-abundance plasma proteins (*Figure 1—figure supplement 1c–f*). The overlap on peptide level corresponded to the predicted pI distribution, with peptides having an isoelectric point covered by several strips being detected repetitively in the corresponding strips (*Figure 1—figure supplement 1g–h*).

In the second step of the optimization, sample loads on the HiRIEF strip were evaluated in terms of analytical depth (*Figure 1b*) and protein identification numbers. This analysis clearly shows a correlation between peptide load on the strip and analytical depth, as well as total numbers of identified proteins when comparing 0.2 mg load to 1 mg, but reaching saturation at 4 mg (*Figure 1c*), a feature which has previously not been shown in cellular/tissue analysis.

To explore the potential of plasma phosphoprotein identification in the acidic HiRIEF range (*Panizza et al., 2017*), we re-searched the data for phosphorylation modifications and found that on average 3.3% of proteins to contain phosphorylated peptides across all strips, including the clinically relevant EGFR. The highest number of phosphopeptides was found in strips covering the most acidic pI range (*Supplementary file 2*). Somewhat surprisingly, the distribution of protein phosphorylation sites differed from data from previous intracellular studies, showing a relatively high proportion of phospho-tyrosine and phospho-threonine modifications (*Lundby et al., 2012*; *Olsen et al., 2006*; *Sharma et al., 2014*) (*Supplementary file 3*).

When summarizing the protein identifications from the different loads and pI intervals in total 3053 proteins were identified in the optimization experiments (*Supplementary file 4*). All raw data as well as result files (protein, peptide and psm tables) from the MS analysis are available in the public repository ProteomeXchange as described in the Materials and methods section.

A comparison of our data to the 3509 proteins from the most recent version of the human plasma PeptideAtlas data set (*Schwenk et al., 2017*) shows that by using different pH intervals, plasma HiRIEF has the ability to cover at least 2236 of the 3509 proteins reported in the PeptideAtlas using gene centric comparison (*Figure 1—figure supplement 2*). Among the 611 plasma proteins detected exclusively by the HiRIEF approach, Golgi membrane proteins and MHC molecules were enriched (*Supplementary file 4*). This shows that the plasma HiRIEF method has the potential to both confirm both previously described plasma proteins and add novel components towards a more complete definition of the plasma proteome.

## Performance assessment of the plasma hirief method

To evaluate the sensitivity of the HiRIEF method, prostate-specific antigen protein (PSA) was spiked-in at a clinically relevant cut-off level of 4 ng/mL into the female plasma and analyzed on the ultranarrow, narrow and broad pH intervals described in the optimization. Interestingly, PSA was only detected in the 4.0–4.25 strip and not in the broader 3.7–4.9 or 3–10 strips, despite covering the same pI interval. This implies that specific, narrower pH intervals could be used to increase analytical depth for selected proteins. In line with this, analyzing the proportion of extracellular, membrane, cytoplasmic and nuclear proteins detected in the different pH intervals, a slightly higher proportion of intracellular proteins were detected in the ultranarrow ranges (*Figure 1—figure supplement 3*).

Another advantage with the HiRIEF methodology is the predictability of the peptide pI (*Zhu et al., 2018*; *Branca et al., 2014*), which allows to design the fraction windows in concordance with the targets of interest. As a proof of concept, we tailored an MRM analysis of fractions 30–35 from the 4.0–4.25 strip that would theoretically contain the PSA peptide R.LSEPAELTDAVK.V and analyzed the selected fractions from the spike-in experiment using MRM analysis. In the experiment, we detected $670 \times 10^{-18}$ moles of the PSA peptide, demonstrating that findings from the global analysis can be validated using targeted MS-methods by rational selection of HiRIEF fractions based on theoretical peptide pI (*Figure 1—figure supplement 4a–b*).

A key feature in plasma proteomics analysis is the ability to robustly detect and quantify proteins across samples in larger cohorts. For this purpose, we combined the HiRIEF methodology with TMT-10 plex labeling and analyzed two separate plasma cohorts collected at different hospitals, with four and five TMT-sets, respectively.

To investigate the repeatability (protein overlap between runs) and quantitative reproducibility of the method we focused on the broad range strip (pH 3–10), as it would generate the highest number of identified peptides which, in turn, would give the best protein quantification accuracy. Due to

the larger pI interval per fraction, the broad range strips give higher peptide focusing accuracy on strips (*Figure 1—figure supplement 5*), which is beneficial for the repeatability.

On each strip, we loaded in total 1 mg of depleted TMT-labeled plasma (100 μg/TMT-label), which was in coherence with the yield from a single 40 μl crude plasma injection per sample on the depletion system (*Figure 1b*).

The first cohort contained samples from 30 healthy individuals, 15 men and 15 women. To connect the quantitative information between the sets and enable reproducibility calculations we created a pooled internal standard of depleted and digested plasma from all 30 individuals, which we included in the study. This allowed us to study the peptide level variability by including replicates of the pooled internal standards in each set - four TMT sets, 30 individual samples and 10 pooled internal standards (*Figure 1—figure supplement 6a*). We identified in total 2587 unique proteins across the TMT-sets, of which 1313 (51%) were detected in all four sets (*Figure 1—figure supplement 6b*). Quantitative reproducibility was calculated both between and within TMT-sets based on the replicates of the pooled internal standard. The average peptide level technical CV (coefficient of variation) within one TMT set was 4.7% and the average CV between sets was of 7.3% (*Figure 1—figure supplement 6c,d*).

In the second cohort, we wanted to explore if we could reduce the MS analysis time, and still retain the proteome coverage. For this purpose, we tailored a condensed 3–10 HiRIEF LC MS/MS analysis approach where fractions in pI areas containing fewer peptides where pooled and analyzed together in the LC-MS/MS analysis. This reduced the number of fractions analyzed from 72 to 40, and the MS analysis time by approximately 30 hr to 55 hr per HiRIEF plate. To test this condensed approach, we analyzed a second longitudinal cohort with plasma samples from 12 women making a total of five TMT-sets. In this analysis, we identified in total 2123 proteins, with 1135 being detected in in all five sets, showing that the reduced analysis time only had a minor impact on the number of identified proteins (*Figure 1—figure supplement 7a,b,c,d*).

Lastly, we compared the identified proteins from the nine TMT-sets from the two different cohorts to define a core set of proteins that could be robustly detected in plasma using the 3–10 strip. In this analysis, we used a gene-centric approach to avoid variation in protein grouping caused by protein inference. Using this approach, we identified in total 828 genes in all nine TMT sets containing samples from 42 individuals (both men and women) (*Figure 1—figure supplement 7e*).

To benchmark the performance of plasma HiRIEF with current state-of-the-art methodologies, we downloaded the available iTRAQ-4 plex, TMT-6 plex and TMT-10 plex datasets from Keshishian *et al.* (*Keshishian et al., 2015*; *Keshishian et al., 2017*) and re-searched the raw data using the same search parameters as for the plasma HiRIEF data. Starting from 400 μl of plasma per sample and performing extensive IgY-based depletion in combination with high-pH reversed phase prefractionation into 30 fractions, Keshishian *et al.* report between 2066 and 4836 proteins identified per set, depending on the labelling approach and experiment, using approximately 90 hr of MS-time per set. Based on spiked-in peptides they present median peptide level CVs between 16–24% across different abundance ranges.

Performing the same gene-centric core-set analysis, as described above for the HiRIEF data, on the re-searched data from the four iTRAQ 4-plex sets (four patients) and the iTRAQ 4-plex, TMT-6 plex, TMT-10 plex analysis (one pooled plasma sample), a set of 1394 genes was robustly identified in all seven sets. In comparison to our data, we identified 1043 genes in seven out of the nine sets, with 42 individuals, starting with 40 μl plasma and 14-protein MARS depletion (*Figure 1—figure supplement 7e*). While identifying slightly less proteins, we believe that the HiRIEF approach has an advantage in throughput, robustness and analytical cost, which makes it well suited for larger clinical cohorts.

## Exploring the plasma proteome

Next, we used plasma HiRIEF to explore the overall plasma protein inter-individual variability by analyzing the plasma from the 30 healthy donors (15 men, 15 women). To define which plasma proteins were tightly controlled or highly variable between individuals, we ranked all overlapping quantified proteins based on inter-individual CV (%). When examining the classes of proteins associated with low or high variability by comparative GO enrichment analyses, we found that coagulation- and complement cascade proteins such as complement factor I (CF1) and complement component C6 were tightly regulated, which was in line with previous findings (*Cominetti et al., 2016*). Interestingly,

transmembrane proteins and proteins coupled to receptor activity, such as the cancer-related EGFR and TGFBR3, were also tightly regulated (*Supplementary file 5*).

Large inter-individual variation was observed for lipoproteins and keratins (the latter classified as probable contamination) (*Supplementary file 6*). One example of proteins with large variability between individuals was lipoprotein A (LPA), for which a genetic variation affecting its secretion into

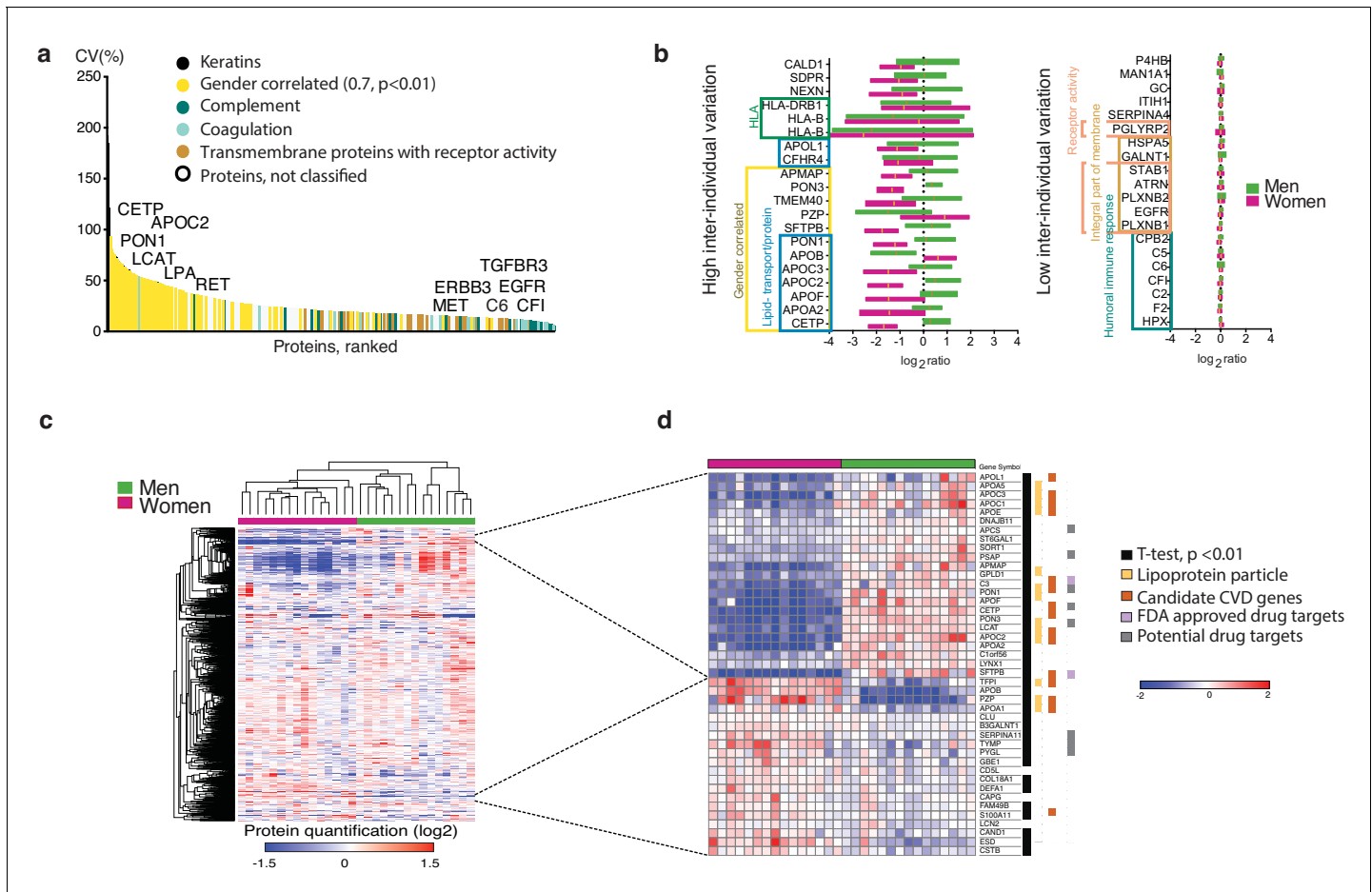

**Figure 2.** Plasma proteome inter-individual variability. (a) The plot shows the coefficient of variation (CV%) of 1080 plasma proteins detected across 30 blood donors, arranged from high to low CV(%). The largest variation was observed among keratins (known common contaminants that were excluded from downstream analyses) and gender-correlated proteins, which by gene ontology (GO) enrichment analyses were identified as enriched for lipoproteins. In contrast, complement and coagulation cascade proteins showed low inter-individual variation. Further analysis of proteins with low inter-individual variation showed enrichment of integral membrane proteins (FDR q-value 7E-11) and receptor activity (FDR q-value, 2E-4), for example the transmembrane receptors EGFR and TGFB3R. The variation did not increase with protein abundance level (*Figure 2—figure supplement 1*). (b) Bar graph showing the 20 proteins with the highest and 20 proteins with the lowest inter-individual variation across all subjects, categorized according to GO annotation and gender-correlated differential expression. All gender-correlated proteins were also statistically-significant when comparing men and women (Student's t-test p<0.01, FDR 1%, not assuming equal variance). (c) Heatmap showing unsupervised clustering of all 1080 proteins with overlapping quantification across the 30 plasma samples. The plasma proteins (log$_2$-transformed ratios relative to a pooled internal standard composed of an aliquot of all 30 samples, rows) and the individual samples (columns) were sorted by hierarchical clustering (Pearson correlation, average linkage), with gender appearing to have the strongest impact on the upcoming clusters. In (d) is shown a zoom-in of the gender-correlated clusters (selection indicated by dashed lines). Indicated are lipoproteins, cardiovascular disease (CVD) genes, and potential- as well as FDA-approved drug targets. Indicated are also proteins with a statistically significant gender-specific differential expression.

DOI: https://doi.org/10.7554/eLife.41608.012

The following figure supplements are available for figure 2:

**Figure supplement 1.** Inter-individual plasma protein abundance variability calculated from plasma samples from 30 donors.
DOI: https://doi.org/10.7554/eLife.41608.013
**Figure supplement 2.** Plasma proteome overview across 30 individuals (15 female and 15 male donors).
DOI: https://doi.org/10.7554/eLife.41608.014

plasma is known (*Boerwinkle et al., 1989*; *Utermann, 1989*) (*Figure 2a*). The HLA genes are similarly known to be highly polymorphic and showed very high level of variability, which could be due to the fact that the peptides were not properly assigned to the reference sequences in the database search.

Two other highly variable proteins (APOC2 and CETP) showed strong gender correlation, with lower concentrations in women (*Figure 2b*). Increased levels of CETP in women with Type two diabetes, but not in men, has been coupled to cardiovascular disease (*Alssema et al., 2007*).

Unsupervised clustering of the data showed distinct gender effects in the plasma proteome (*Figure 2c*), that are concordant with previous studies (*Corzett et al., 2010*; *Miike et al., 2010*). Interestingly, the major gender-correlated cluster with relative up-regulation in male subjects contained proteins linked to lipid transport and binding. In fact, 15 of the 38 proteins in the cluster with significant gender-specific differential expression (Student's t-test, p<0.01, FDR corrected) were proteins implicated as candidate cardiovascular disease genes, including seven potential drug targets and two FDA-approved drug targets (classification according to the ProteinAtlas, www.proteinatlas.org) (*Figure 2d*, *Figure 2—figure supplement 2*).

## Tissue annotated proteins detected in plasma

To explore the potential of detecting biomarkers using plasma HiRIEF, we mined our data for the proteins included in the recently published CancerSEEK test (*Cohen et al., 2018*). We identified five out of eight proteins in the optimization experiments (covering several pI intervals) and three of eight proteins in the cohort consisting of 30 healthy plasma donors analyzed only on the 3–10 strip (*Figure 3a*, *Supplementary file 7*).

We then expanded the criteria and searched for FDA-approved drug targets, cancer-related proteins and possible tissue leakage proteins, all classified according to the Human Protein Atlas (HPA) (*Uhlén et al., 2015*). The latter defined by using the most stringent human tissue proteome annotations provided by HPA - 'tissue enriched'. We ranked the merged data from the optimization based on abundance and colored the proteins according to protein classes (*Figure 3b*). As expected, classical plasma proteins were detected in the high-abundance range and potential biomarker classes (FDA-approved drug targets, cancer-related proteins, the proteins from the CancerSEEK test and the spiked PSA) were detected in the lower abundance ranges (*Figure 3b*, *Figure 3—figure supplement 1*, and *Supplementary file 8*). Notably, using plasma HiRIEF, we could detect over 500 tissue-enriched proteins spanning over a wide range of concentrations, from highly abundant plasma proteins produced in the liver to less abundant proteins produced in the central nervous system or the pancreas (*Figure 3c*, *Figure 3—figure supplement 1*, and *Supplementary file 8*).

As a proof of concept to demonstrate the detection of potential tissue leakage proteins using the HiRIEF method, we searched for the presence of placenta-enriched proteins in the cohort with 30 blood donors (non-pregnant, both men and women) as we would not expect to detect any placenta proteins in these individuals. In this cohort, we detected low levels of eight proteins classified as placental enriched (median #PSMs 2, range 0–21, the outlier protein IGF2; *Figure 4a*).

We then obtained and analyzed plasma from a healthy female donor before pregnancy and during third trimester of pregnancy, again on the 3–10 strip as for the healthy donors. During pregnancy, we could detect 30 tissue enriched placental proteins in plasma, with median # of PSM´s 46 (range 1–424), which were not detected, or detected at low levels prior to pregnancy (median #of PSMs 0, range 0–31, again the outlier protein being IGF2) (*Figure 4b*).

We continued the analysis to search for increased amounts of placental proteins in newborns, as the placenta is mainly of fetal origin. Two mother and newborn pairs were analyzed using plasma HiRIEF on HiRIEF 3–10 strips. In this cohort we found over 30 placenta-enriched proteins at high levels in the maternal plasma (*Figure 4c*), but only a small increase of placental proteins among the newborns, with a median PSM ratio mother:newborn of 4:1 and 5:1, respectively.

Expanding the comparison between newborn and maternal plasma to the full plasma proteome, we compared protein abundances of mother- versus baby-proteomes and also the pregnant- versus not-pregnant proteomes of the female donor by plotting the protein abundances against each other (*Figure 4d*). Among the proteins uniquely detected in both babies were Anti-Mullerian-Hormone (AMH), which has reported serum level concentrations of 52 ng/mL in newborn boys (*Bergadá et al., 2006*) (both newborns were boys) compared to approximately 2 ng/mL in females during pregnancy and puerperium (*La Marca et al., 2005*). Among the mother-unique proteins was

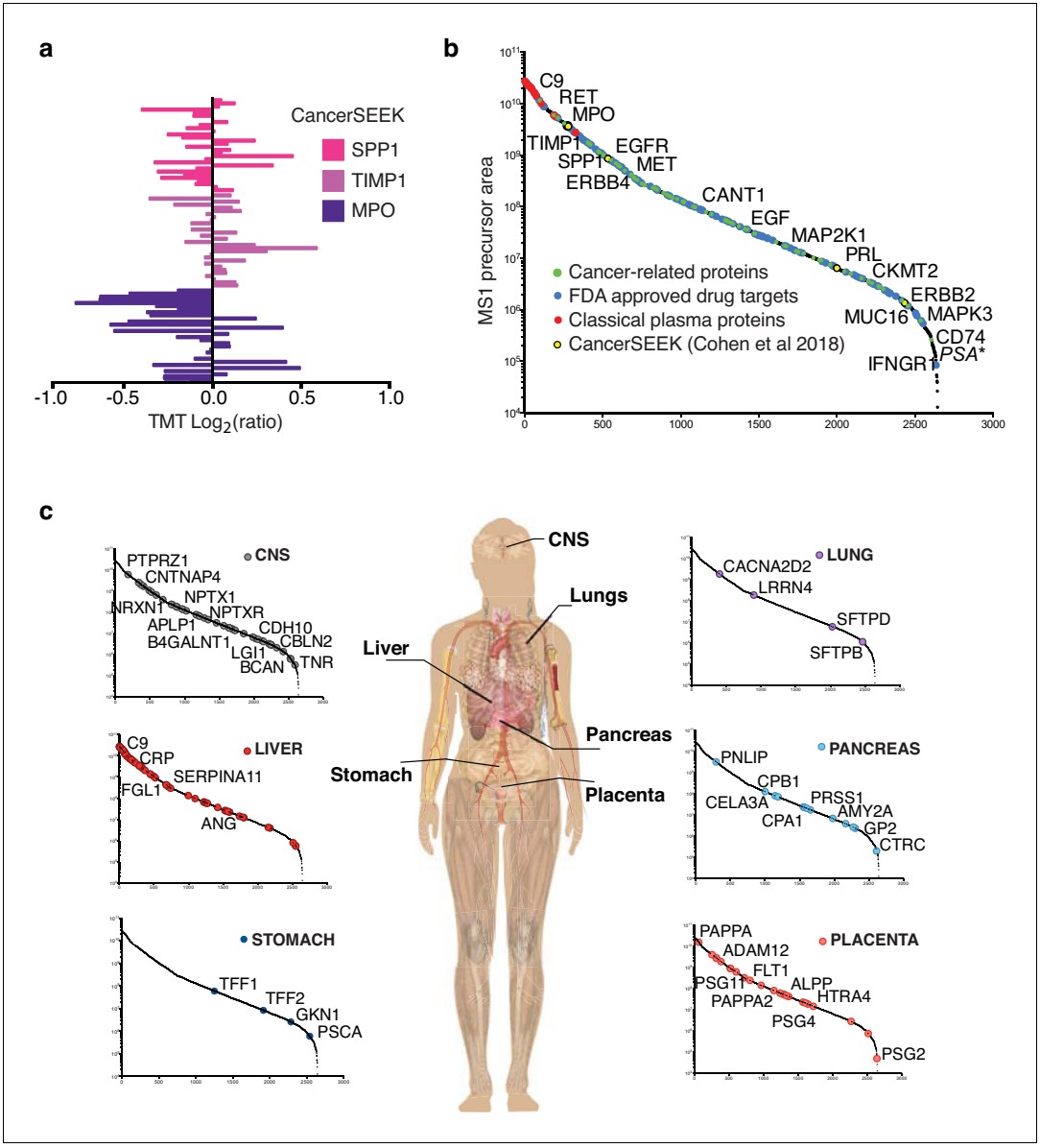

**Figure 3.** Exploring the plasma proteome and detecting tissue leakage proteins in plasma. (**a**) Individual protein expression levels (log₂(relative ratios)) of the three out of eight proteins of the CancerSEEK test (*Cohen et al., 2018*) that were detected in a cohort of 30 blood donors. (**b**) Distribution of protein abundances based on precursor areas. Cancer-related proteins and FDA-approved drug target proteins (HPA classification) highlighted in green and blue, respectively. Classical plasma proteins (*Anderson and Anderson, 2002*) highlighted in red, proteins included in the CancerSEEK test in yellow. Among these proteins, some are specifically indicated by gene symbol. Spiked PSA (4 ng/mL) is marked with *. (**c**) Detection of tissue leakage proteins in plasma assessed by mRNA expression levels classified as enriched on particular tissue types. The anatomy image in panel c was adapted from https://commons.wikimedia.org/wiki/File:Female_shadow_anatomy_without_labels.png, which is available in the Public Domain.

DOI: https://doi.org/10.7554/eLife.41608.015

The following figure supplement is available for figure 3:

**Figure supplement 1.** Scatter-dot plot of the merged data from the optimization, with the proteins subdivided into classes: classical plasma proteins (*Eden et al., 2007*), HPA classification of 1) FDA-approved drug target proteins, 2) cancer-related proteins and 3) tissue leakage proteins in plasma (assessed by mRNA expression levels classified as enriched on particular tissue types: shown is liver, CNS, pancreas) and finally proteins included in the CancerSEEK test (*Cohen et al., 2018*).

DOI: https://doi.org/10.7554/eLife.41608.016

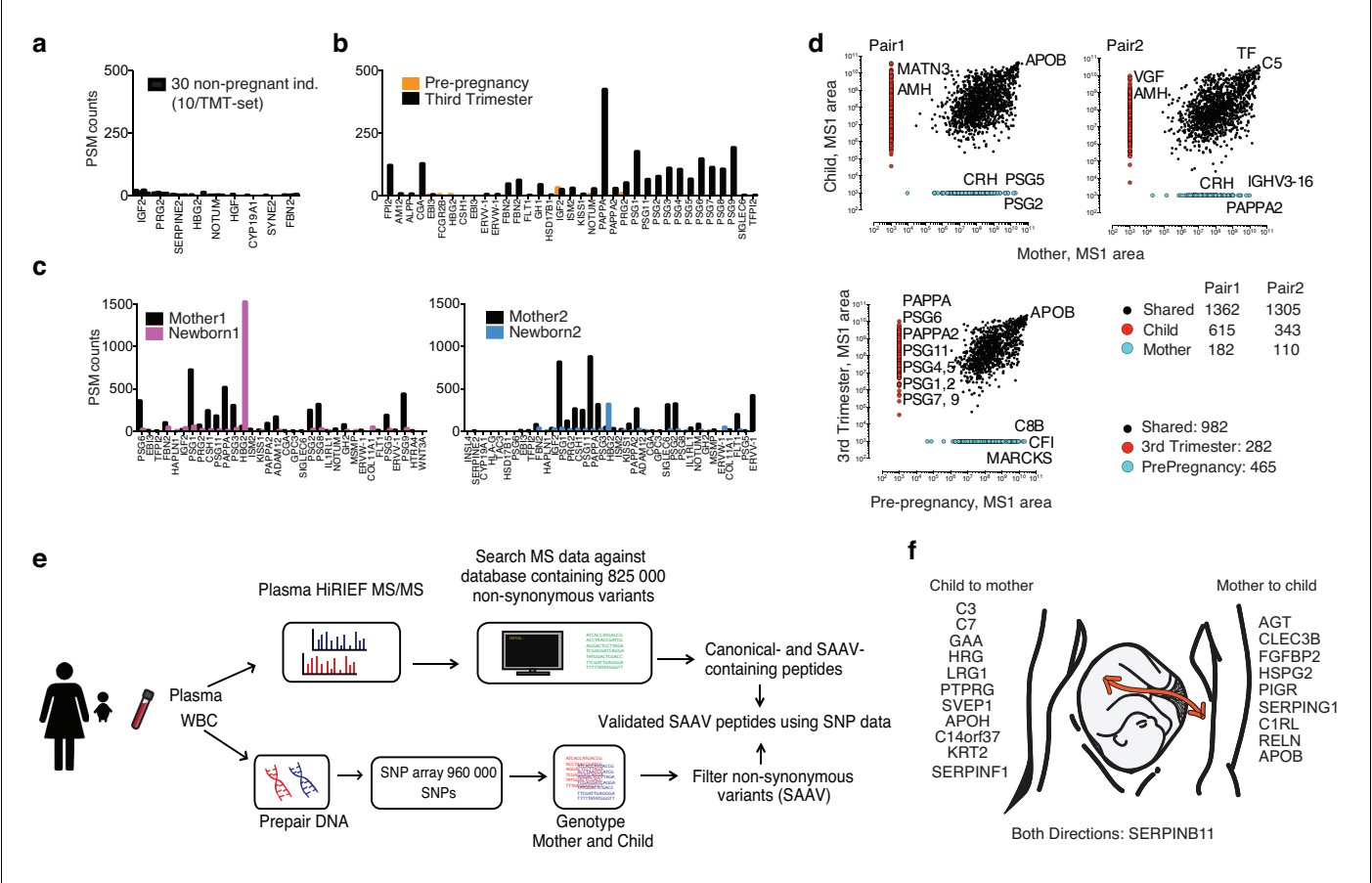

**Figure 4.** Use of theoretical peptide pI in global and targeted plasma HiRIEF. (**a**) Placental proteins detected in 30 non-pregnant blood donors. (**b**) Placental proteins detected at two different time points in one individual, during third trimester and prior to pregnancy. (**c**) Placental proteins in two pairs of mother and newborn. (**d**) A comparison of unique and shared plasma protein levels (MS1 area) in mother versus newborn (two pairs), and in one female individual pre-pregnancy versus third trimester. (**e**) A brief overview of the proteogenomics workflow (**f**) An illustration of suggested placental leakage proteins and direction of transfer between mother-and-child.

DOI: https://doi.org/10.7554/eLife.41608.017

The following figure supplement is available for figure 4:

**Figure supplement 1.** Protein abundance distribution of the 24 proteins possibly transferred across the placenta between mother and child and vice versa.

DOI: https://doi.org/10.7554/eLife.41608.018

corticotropin-releasing hormone (CRH), a protein suggested as a trigger for parturition due to its rapid increase in circulating levels at the onset of parturition (*Zannas and Chrousos, 2015*). Among the pregnancy-unique proteins were numerous placenta-enriched proteins (PAPPA2, PSGs).

Fetal hemoglobin was found at very high levels in the newborns (1501 and 317 PSMs, respectively), as expected, but a fraction of fetal hemoglobin was also detected in the mother's blood (55 and 44 PSMs, respectively). The fetal-specific isoform of haemoglobin has previously been detected maternal plasma during pregnancy and transfer across the placenta has been proposed, however, it has also been shown that the level of fetal hemoglobin expressed in adults has a genetic component (*Thein et al., 2007*; *Galarneau et al., 2010*), making it difficult to determine if it is produced by the mother or transferred across the placenta.

Fetal DNA can easily be detected in maternal plasma and is transferred across the placenta during pregnancy (*Lo et al., 2007*). Several proteins are also suggested to pass the placenta (*Wölter et al., 2016*), but providing evidence that such proteins have not been produced in the recipient has remained an analytical challenge.

## Plasma proteogenomics

To pinpoint proteins that could have been transferred across the placenta, we adopted a proteogenomics approach (*Figure 4e*). First, we wanted to know if we could detect single amino acid variants (SAAVs) in proteins in plasma that could be used as a traceable protein fingerprint, since this has not been previously shown in plasma. We therefore collected plasma and buffy coat from two mother-newborn pairs in connection to planned Caesarean section. The buffy coats containing white blood cells (WBC) were genotyped using SNP arrays and the corresponding plasma samples were analyzed by plasma HiRIEF (3–10 strip) to detect SAAV using a customized database including all coding SNPs in dbSNP (database generation as described in detail by *Zhu et al., 2018*).

The MS data was subsequently curated using SpectrumAI within the iPAW pipeline to reduce false discoveries among the detected single amino acid substitutions (*Zhu et al., 2018*). Indeed, in the plasma samples from the four individuals, we detected 229 peptides with unique SAAV (mapping to in total 161 different genes), where the corresponding SNPs were also detected on DNA level in the SNP analysis (*Supplementary file 9*). In the cases where we detected a SAAV-containing peptide and the individual was heterozygous for the SNP, we also found the corresponding canonical peptide in 65% of the cases. These findings strengthened the hypothesis that SAAVs could be used as traceable protein fingerprints detected in plasma.

We then explored if the curated SAAV peptides detected by MS that could not be explained by the individual's own genotype could possibly be originating from a transfer across the placenta. Indeed, from the two mother-child pairs we found 32 cases with in total 26 unique peptides sequences from 24 proteins where the amino acid sequence could be explained by the genotype of other the individual, indicating a possible transfer of these proteins across the placenta. Mirror plots of spectra from the peptide in the donor (with DNA support) and the recipient (without DNA support) can be found in *Supplementary file 13*. After manual inspection of the mirror plots, 5 out of the 32 cases where considered poor matches, leaving 21 proteins and 22 unique peptides. The mismatch rate is similar to the rate previously reported using synthetic peptides for sequence verification (*Zhu et al., 2018*).

For the majority of the proteins (n = 11), the transfer was detected in the direction of baby to the mother, but we also detected nine proteins potentially transferred from the mother to the baby. For three of these transferred proteins, we could see the same transfer directions in both of the mother-child pairs, strengthening our findings (GAA from baby to mother, LRG from baby to mother and HSPG2 from mother to baby). For one additional transferred proteins, we saw opposing transfer directions in the two pairs, suggesting that the protein could transfer in both directions (SERPINB11) (*Figure 4f*).

The transferred proteins were spread across the entire abundance range and did not stand out in terms of hydrophilicity/hydrophobicity (as evaluated by Gravy score), size, pI or abundance (*Figure 4—figure supplement 1* and *Supplementary file 10*). A detailed description of the transferred proteins, including peptides and PSMs can be found in *Supplementary file 11*.

To our knowledge, this is the first-time plasma proteogenomics has been used to study protein transfer across the placenta in-vivo providing a new possibility to study the molecular communication between mother and baby.

## Discussion

In the last 5 years, the MS-based proteomics field has taken major steps forward in terms of proteome coverage and analytical depth. Identification and quantification of over 10 000 proteins can now routinely be performed in cell line and tissue samples, which is in line with the number of transcripts reported using RNA sequencing from same samples (*Thul et al., 2017*). Global proteomics analysis of plasma on the other hand still remains a challenge. Compared to cellular and tissue analysis, plasma proteomics has in particular high demands on throughput and dynamic range. Inherent to plasma analysis is also a very high sample-to-sample variability and lack of genetic blueprint specific for plasma. Taken together, these challenges call for specific adaptions of MS based proteomics analysis for plasma.

In their review 'Revisiting biomarker discovery by plasma proteomics' Geyer *et al* argues for a rectangular biomarker strategy, where both the discovery and validation cohort are analyzed using global MS-based proteomics (*Geyer et al., 2017*). This is an appealing approach, as it enables

seamless validation of protein species that would be difficult to validate using other technologies, such as proteins containing SAAV or larger panels of proteins. In the current paper, our aim has been to balance the demand on throughput and analytical depth, to develop method based on HiR-IEF-MS that performs robustly in an in-depth plasma proteome analysis but also at same time has the capability to analyze larger clinical cohorts.

We show that that the method has the capability to repeatedly detect tissue leakage proteins in plasma and that the robust high-precision fractionation enables quantification using several TMT-sets. We also show the method's potential to detect phosphoproteins and single amino acid variant peptides adding a proteome dimension in to plasma analysis that is difficult to cover by antibodies due to lack of specific reagents.

Lastly, we highlight the potential of HiRIEF-MS as a tool for plasma proteogenomics by showing its capability of deciphering the transfer of protein variants between mother and child during pregnancy. The approach provides new and unique possibilities for studies on signaling between the mother and the fetus as well as a great possibility for improving our understanding of pathological pregnancies.

## Materials and methods

### Plasma collection

#### Plasma sampling

Peripheral venous blood from a healthy female donor was collected in EDTA tubes (BD Vacutainer K2E 7.2 mg, BD Diagnostics) and kept on ice until preparation, to prevent coagulation and minimize protein degradation. EDTA tubes were first centrifuged at 1500 × g at 4°C for 10 min. The supernatant was transferred to a new tube and centrifuged at 3000 × g at 4°C for 10 min. The plasma was then aliquoted and kept at −80°C until analysis. All samples used in this study were prepared within 1 hr of sample collection and showed no signs of hemolysis.

#### Optimization cohort

Plasma from an anonymous female donor was collected and pooled for the optimization experiment. Aliquots from the same pooled sample was used for all the experiments in the optimization study to reduce experimental variability.

#### Mother-child cohort

Two healthy pregnant women with planned Caesarian section (due to Tokophobia) were asked to join the study. Plasma was sampled from the mothers prior to C-section, and the cord blood was taken from the child directly after birth. Plasma was prepared as described in previous section and in addition buffy coat, containing white blood cells, was collected after the plasma separated. All donors signed informed consent and the study was approved by the local ethics committee, ethical approval number 2014/1622-31/2.

#### Thirty healthy donors cohort

Healthy men and women were asked to join the study. Ethical permits were obtained from the ethical committee at Karolinska University Hospital (Dnr 91:164 for men and Dnr 95:298 for women.) All participants signed informed consent prior to inclusion in the study.

#### Female longitudinal cohort

Individuals were recruited to a longitudinal study where plasma was collected prior to influenza vaccination (day 0) and at day 1, 28, and 90 days post-vaccination. All donors signed informed consent prior to study inclusion and the study was approved by the local ethics committee, ethical approval number 2008/915-31/4 m.

## Plasma sample preparation

### Depletion of high-abundance proteins

Agilent Plasma 14 Multiple Removal System 4.6 × 100 (Agilent Technologies) was set up on an Agilent HPLC system (Agilent technologies) and run according to the manufacturer's instructions. The depleted plasma flow-through was concentrated on 5 kD molecular weight cut off filter (Agilent Technologies) followed by buffer exchange >100 times to 20 mM Ammonium bicarbonate for the optimization study and 50 mM HEPES for the TMT-analyses.

### Digestion and labeling

Depleted plasma was denatured at 95℃ for 10 min followed by reduction with dithiothreitol and alkylation with iodoacetamide at end concentrations 5 mM and 10 mM, respectively. Trypsin (Mass Spec Gold, Promega) was added at a 1:50 (w/w) ratio and digestion was performed at 37℃ overnight. When applicable TMT-labeling was performed according to manufacturer's instructions (Thermo scientific).

Following digestion (and labeling if applicable), 1 ml Strata X-C 33 µm columns (Phenomenex) were used for sample clean-up. The peptides were subsequently dried in a speedvac.

### HiRIEF separation

Briefly, HiRIEF was performed as previously described (*Branca et al., 2014*). The samples were rehydrated in 8 M urea with bromophenol blue and 1% IPG buffer, and subsequently loaded to the immobilized pH gradient (IPG) strip and run according to previously published isoelectric focusing (IEF) protocols (*Branca et al., 2014*; *Zhu et al., 2018*). After IEF, the IPG strip was eluted into 72 fractions using in-house robot. The obtained fractions were dried using SpeedVac and frozen at −20℃ until MS analysis.

## Plasma optimization study

### Sample preparation

Plasma from healthy female donor with and without PSA spiked-in was depleted and digested as described above. In total, 16 different conditions were evaluated by label-free analysis. A detailed summary of the different pI intervals and sample loads in the HiRIEF optimization can be found in *Supplementary file 2*. To create an MS runtime control, 72 aliquotes of each 1 µg of peptides from depleted plasma was analyzed using identical LC-MS/MS settings as for the HiRIEF analysis.

### LC-ESI-MS/MS Q-Exactive

Online LC-MS was performed using a hybrid Q-Exactive mass spectrometer (Thermo Scientific). For each LC-MS/MS run, the auto sampler (Dionex UltiMate 3000 RSLCnano System) dispensed 15 µl of mobile phase A (3% acetonitrile, 0.1% formic acid) to the well of the microtiter plate (96 well V-bottom, polypropylene, Greiner), mixed 10 times (by repeatedly aspirating and dispensing 10 µl), and proceeded to inject 7 µl into a C18 guard desalting column (Acclaim pepmap 100, 75 µm x 2 cm, nanoViper, Thermo). After 5 min of flow at 5 µl/min with the loading pump, the 10-port valve switched to analysis mode in which the NC pump provided a flow of 250 nL/min through the guard column. The curved gradient (curve six in the Chromeleon software) then proceeded from 3% mobile phase B (95% acetonitrile, 5% water, 0.1% formic acid) to 45% B in 50 min followed by wash at 99% B and re-equilibration. Total LC-MS run time was 74 min. We used a nano EASY-Spray column (pepmap RSLC, C18, 2 µm bead size, 100 Å, 75 µm internal diameter, 50 cm long, Thermo) on the nano electrospray ionization (NSI) EASY-Spray source (Thermo) at 60℃.

FTMS master scans with 70,000 resolution (and mass range 300–1700 m/z) were followed by data-dependent MS/MS (35 000 resolution) on the top five ions using higher energy collision dissociation (HCD) at 30–40% normalized collision energy. Precursors were isolated with a 2 m/z window. Automatic gain control (AGC) targets were 1e6 for MS1 and 1e5 for MS2. Maximum injection times were 100 ms for MS1 and 450 ms for MS2. The entire duty cycle lasted ~3.5 s. Dynamic exclusion was used with 60 s duration. Precursors with unassigned charge state or charge state one were excluded. An underfill ratio of 0.1% was used.

## Data searches

Raw MS/MS files were converted to mzML format using msconvert from the ProteoWizard tool suite (*Kessner et al., 2008*). Spectra were then searched in the Galaxy framework using tools from the Galaxy-P project (*Goecks et al., 2010*; *Boekel et al., 2015*), including MSGF+ (*Kim and Pevzner, 2014*) (v10072) and Percolator (*Käll et al., 2007*) (v2.10), where eight subsequent HiRIEF search result fractions were grouped for Percolator target/decoy analysis. Peptide and PSM FDR were recalculated after merging the percolator groups of eight search results into one result per HiRIEF analysis. The reference database used was the human protein subset of ENSEMBL 80. Both gene-centric and protein centric protein grouping were performed. Quantification was performed as follows: For label-free quantification, protein MS1 precursor area was calculated as the average of the top three most intense peptides, and peptide MS1 precursor area as the highest PSM area for each peptide. Inferred gene identity false discovery rates were calculated using the picked-FDR method (*Savitski et al., 2015*), whereas the FDR for protein level identities was calculated using the -log10 of best-peptide q-value as a score. The search settings included enzymatic cleavage of proteins to peptides, using trypsin limited to fully tryptic peptides. Carbamidomethylation of cysteine was specified as a fixed modification. The minimum peptide length was specified to be six amino acids. Variable modifications (maximum four allowed) were oxidation (of methionine) and phosphorylation (of serine-threonine-tyrosine). Searches not including phosphorylation as variable modification were also done.

MS data for the optimization study was also analyzed using the MaxQuant software package (version 1.5.3.30). The Andromeda search engine (*Cox and Mann, 2008*) was used to search the MS/MS spectra against the human ENSEMBL database (v80) to identify corresponding proteins. Default parameters were used except for protein identification and quantification, which was limited to unique peptides. In brief, the false-discovery rate (FDR) was fixed to a threshold of 1% FDR at PSM and protein level. FDR was estimated using a target-decoy database search approach. The minimum peptide length was specified to be six amino acids. Carbamidomethylation of cysteine was specified as a fixed modification. Methionine oxidation, N-terminal protein acetylation of lysine and phosphorylation of serine/threonine/tyrosine were chosen as variable modifications. Site FDR < 1%. Data are available via ProteomeXchange with identifier PXD010899.

## PSA Spike-in

Purified prostate specific antigen (PSA) protein (ab78528, Abcam) was spiked in at 4 ng/mL in crude EDTA plasma from healthy female donor.

## PSA ELISA

To ensure that PSA was not removed during the depletion process, intact PSA protein was measured using ELISA in female plasma spiked-in with PSA at 4 ng/mL, depleted spiked-in plasma and two samples from patients diagnosed with metastatic prostate cancer (serving as positive controls) using ELISA kit R and D Systems #DKK300.

## PSA LC-MRM analysis

Fraction 30–35 from the HiRIEF 4.0–4.25 strip from the optimization experiment was analyzed using LC-MRM. Total peptide load on the strip was 0.6 mg of depleted female plasma with PSA spiked-in (described above).

A Capillary pump (from an Agilent 1200 LC system) was used to load sample into a C18 guard desalting column (Zorbax 300 SB-C18, 5 × 0.3 mm, 5 μm bead size, Agilent), at a flow rate of 4 μl/min with 100% of loading mobile phase A (3% ACN, 0.1% FA). After loading for 3 min, the 6-port valve switched and the C18 guard desalting column was put in line with the nano pump, and the sample was eluted and separated using a 15 cm long C18 picofrit column (100 μm internal diameter, 5 μm bead size, Nikkyo Technos Co., Tokyo, Japan). Peptide separation was obtained using a 0.4 μl/min flow rate and a gradient from 3% to 100% of analytical mobile phase B over 45 min. Analytical mobile phase A consisted of 5% DMSO, 0.1% FA, whereas mobile phase B consisted of 90% ACN, 5% DMSO, 0.1% FA. The gradient method was linear and consisted in (time: %B): 0 min: 3% B; 3 min: 3% B; 20 min: 40% B; 25 min: 100% B; 33 min: 100% B; 34 min: 3% B; 45 min: 3% B. Eluted peptides were ionized into a nano electrospray ionization source and analyzed using a triple quadrupole

QQQ Agilent 6490 equipped with iFunnel technology. Default 380 V fragmentor voltage and 5 V cell accelerator potential were used for all peptide transitions. All acquisition methods used the following parameters: Capillary voltage of 2100 V, drying gas flow 11 l/min (UHP Nitrogen) at temperature of 250°C, MS operating pressure of $5 \times 10^{-5}$ Torr, Q1 and Q3 resolution was equal at unit value of 0.7 FWHM.

All MRM data were analyzed with Skyline platform and each peak was manually inspected and integrated. Five transitions were used to identify the target peptide and only the area of the most abundant transition was used to quantify the PSA. The concentration of endogenous peptides was calculated by the ratio of heavy area to endogenous area and the known concentration of heavy-spiked-in peptide. Peptide concentration reported in fmol/μl was converted to ng/ml using the molecular weight of the whole protein.

## PredpI algorithm
To predict pI of identified peptides during the optimization, we used the Pred pI Algorithm available as Supplementary software in *Branca et al. (2014)*. The PredpI algorithm builds on a pI prediction algorithm used for proteins (*Bjellqvist et al., 1993*) that is currently the basis of the Compute pI tool in Expasy (http://web.expasy.org/compute_pi/).

## Reproducibility: healthy donors study
### Sample preparation
Thirty plasma samples were depleted, digested and TMT-labeled as described above. An internal standard was created by pooling an aliquote from each of the 30 samples, in order to connect the four TMT sets together and to enable reproducibility. An overview of the TMT-labelling scheme can be found in *Figure 1—figure supplement 6a*. In total, 1 mg of peptide samples were loaded per pH 3–10 strip and separated as described in the HiRIEF section above.

### LC-ESI-MS/MS Q-Exactive
Online LC-MS was performed using a hybrid Q-Exactive mass spectrometer (Thermo Scientific). For each LC-MS/MS run, the auto sampler (Dionex UltiMate 3000 RSLCnano System) dispensed 15 μl of mobile phase A (3% acetonitrile, 0.1% formic acid) to the well of the microtiter plate (96 well V-bottom, polypropylene, Greiner), mixed 10 times (by repeatedly aspirating and dispensing 10 μl), and proceeded to inject 7 μl into a C18 guard desalting column (Acclaim pepmap 100, 75 μm x 2 cm, nanoViper, Thermo). After 5 min of flow at 5 μl/min with the loading pump, the 10-port valve switched to analysis mode in which the NC pump provided a flow of 250 nL/min through the guard column. The curved gradient (curve six in the Chromeleon software) then proceeded from 3% mobile phase B (95% acetonitrile, 5% water, 0.1% formic acid) to 45% B in 50 min followed by wash at 99% B and re-equilibration. Total LC-MS run time was 74 min. We used a nano EASY-Spray column (pepmap RSLC, C18, 2 μm bead size, 100 Å, 75 μm internal diameter, 50 cm long, Thermo) on the nano electrospray ionization (NSI) EASY-Spray source (Thermo) at 60°C.

FTMS master scans with 70,000 resolution (and mass range 300–1700 m/z) were followed by data-dependent MS/MS (35 000 resolution) on the top five ions using higher energy collision dissociation (HCD) at 30–40% normalized collision energy. Precursors were isolated with a 2 m/z window. Automatic gain control (AGC) targets were 1e6 for MS1 and 1e5 for MS2. Maximum injection times were 100 ms for MS1 and MS2 150. The entire duty cycle lasted ~3.5 s. Dynamic exclusion was used with 60 s duration. Precursors with unassigned charge state or charge state one were excluded. An underfill ratio of 0.1% was used.

### Data searches
Raw MS/MS files were converted to mzML format using msconvert from the ProteoWizard tool suite (*Kessner et al., 2008*). Spectra were then searched in the Galaxy framework using tools from the Galaxy-P project (*Goecks et al., 2010*; *Boekel et al., 2015*) including MSGF+ (*Kim and Pevzner, 2014*) (v10072) and Percolator (*Käll et al., 2007*) (v2.10), where eight subsequent HiRIEF search result fractions were grouped for Percolator target/decoy analysis. Peptide and PSM FDR were recalculated after merging the percolator groups of eight search results into one result per TMT set. The reference database used was the human protein subset of ENSEMBL 80. Quantification of isobaric

reporter ions was done using OpenMS project's IsobaricAnalyzer (*Röst et al., 2016*) (v2.0). Both gene centric and protein centric protein groupings were performed. Quantification on TMT reporter ions in MS2 was for both protein and peptide level quantification based on median of PSM ratios, limited to PSMs mapping only to one protein and with an FDR q-value <0.01. Inferred gene identity false discovery rates were calculated using the picked-FDR method (*Savitski et al., 2015*), whereas the FDR for protein level identities was calculated using the -log10 of best-peptide q-value as a score. The search settings included enzymatic cleavage of proteins to peptides using trypsin, limited to fully tryptic peptides. Carbamidomethylation of cysteine was specified as a fixed modification. The minimum peptide length was specified to be six amino acids. Variable modifications (maximum four allowed) were oxidation (of methionine) and phosphorylation (of serine-threonine-tyrosine). Searches not including phosphorylation as a variable modification were also done. Data are available via ProteomeXchange with identifier PXD010899.

## Reproducibility: female longitudinal

### Sample preparation
Plasma samples were depleted, digested and TMT-labeled as described above. An internal standard was created to connect the five TMT sets together. An overview of the TMT-labeling scheme can be found in *Figure 1—figure supplement 7a*. In total 1 mg of peptide sample were loaded per pH 3– 10 strip and separated as described in the HiRIEF section above. Pooling scheme for the condensed HiRIEF analysis can be found in *Supplementary file 12*.

### LC-ESI-MS/MS Q-Exactive
Online LC-MS was performed using a Dionex UltiMate 3000 RSLCnano System coupled to a Q-Exactive mass spectrometer (Thermo Scientific). Each plate well was dissolved in 20 µl solvent A and 10 µl were injected. Samples were trapped on a C18 guard-desalting column (Acclaim PepMap 100, 75 µm x 2 cm, nanoViper, C18, 5 µm, 100 Å), and separated on a 50cm-long-C18 column (Easy spray PepMap RSLC, C18, 2 µm, 100 Å, 75 µm x 50 cm). The nano capillary solvent A was 95% water, 5% DMSO, 0.1% formic acid; and solvent B was 5% water, 5% DMSO, 90% acetonitrile, 0.1% formic acid. At a constant flow of 0.25 µl min$^{-1}$, the curved gradient (curve six in the Chromeleon software, running under Xcalibur) went from 2% B up to 40% B in each fraction as shown in *Supplementary file 12*, followed by a steep increase to 100% B in 5 min, and subsequent re-equilibration with 2% B.

FTMS master scans with 70,000 resolution (and mass range 300–1600 m/z) were followed by data-dependent MS/MS (35 000 resolution) on the top five ions using higher energy collision dissociation (HCD) at 30% normalized collision energy. Precursors were isolated with a 2 m/z window and an offset of 0.5 m/z. Automatic gain control (AGC) targets were 1e6 for MS1 and 1e5 for MS2. Maximum injection times were 100 ms for MS1 and 450 ms for MS2. Dynamic exclusion was used with 30 s duration. Precursors with unassigned charge state or charge state 1, 7, 8, >8 were excluded.

### Data searches
Raw MS/MS files were converted to mzML format and corrected for mass shifts using the msconvert tool from the ProteoWizard suite (*Keshishian et al., 2007*). Spectra were then searched by MSGF+ (*Kim and Pevzner, 2014*), and post processed with Percolator (*Käll et al., 2007*) in a Nextflow pipeline (*Di Tommaso et al., 2017*), using a concatenated target-decoy strategy.

The reference databases used were the human protein database of Swissprot at 2018-07-18. MSGF +settings included precursor mass tolerance of 10ppm, fully-tryptic peptides, maximum peptide length of 50 amino acids and a maximum charge of 6. Fixed modifications were TMT-10-plex on lysines and N-termini, and carbamidomethylation on cysteine residues, a variable modification was used for oxidation on methionine residues. Quantification of TMT-10plex reporter ions was done using OpenMS project's IsobaricAnalyzer (*Röst et al., 2016*) (v2.0). PSMs found at 1% PSM- and peptide-level FDR (false discovery rate) were used to infer gene identities, which were quantified using the medians of per-channel median-normalized PSM quantification ratios. Inferred gene identity false discovery rates were calculated using the picked-FDR method, whereas the FDR for protein level identities was calculated using the –log10 of best-peptide q-value as a score (*Savitski et al., 2015*).

The full pipeline (version ee45bf6) is available at https://github.com/lehtiolab/galaxy-workflows/ (copy archived at https://github.com/elifesciences-publications/galaxy-workflows; *Boekel, 2019*).

## Proteogenomics analysis of mother and child cohort

### Sample preparation

Plasma samples from two mothers and two babies were subjected to depletion as described above. In this analysis, both the depleted fraction and the proteins bound to the column were analyzed using HiRIEF 3–10 strips as described in the HiRIEF section above. The bound fraction was included because we did not want to exclude any of the high-abundance proteins in the transfer analysis. In total 1 mg of digested peptides were loaded on each strip and analyzed with label-free mass spectrometry.

### LC-ESI-MS/MS Q-Exactive

Online LC-MS was performed using a hybrid Q-Exactive mass spectrometer (Thermo Scientific). For each LC-MS/MS run, the auto sampler (Dionex UltiMate 3000 RSLCnano System) dispensed 15 µl of mobile phase A (3% acetonitrile, 0.1% formic acid) to the well of the microtiter plate (96-well V-bottom, polypropylene, Greiner), mixed 10 times (by repeatedly aspirating and dispensing 10 µl), and proceeded to inject 7 µl into a C18 guard desalting column (Acclaim pepmap 100, 75 µm x 2 cm, nanoViper, Thermo). After 5 min of flow at 5 µl/min with the loading pump, the 10-port valve switched to analysis mode in which the NC pump provided a flow of 250 nL/min through the guard column. The curved gradient (curve six in the Chromeleon software) then proceeded from 3% mobile phase B (95% acetonitrile, 5% water, 0.1% formic acid) to 45% B in 50 min followed by wash at 99% B and re-equilibration. Total LC-MS run time was 74 min. We used a nano EASY-Spray column (pepmap RSLC, C18, 2 µm bead size, 100 Å, 75 µm internal diameter, 50 cm long, Thermo) on the nano electrospray ionization (NSI) EASY-Spray source (Thermo) at 60°C.

FTMS master scans with 70,000 resolution (and mass range 300–1700 m/z) were followed by data-dependent MS/MS (35,000 resolution) on the top five ions using higher energy collision dissociation (HCD) at 30–40% normalized collision energy. Precursors were isolated with a 2 m/z window. Automatic gain control (AGC) targets were 1e6 for MS1 and 1e5 for MS2. Maximum injection times were 100 ms for MS1 and 450 ms for MS2. The entire duty cycle lasted ~3.5 s. Dynamic exclusion was used with 60 s duration. Precursors with unassigned charge state or charge state one were excluded. An underfill ratio of 0.1% was used.

### Data searches

Raw MS/MS files were converted to mzML format using msconvert from the ProteoWizard tool suite (*Kessner et al., 2008*). Spectra were then searched in the Galaxy framework using tools from the Galaxy-P project (*Goecks et al., 2010*; *Boekel et al., 2015*), including MSGF+ (*Kim and Pevzner, 2014*) (v10072) and Percolator (*Käll et al., 2007*) (v2.10), where eight subsequent HiRIEF search result fractions were grouped for Percolator target/decoy analysis. Peptide and PSM FDR were recalculated after merging the percolator groups of eight search results into one result per HiRIEF analysis. The reference database used was the human protein subset of ENSEMBL 80. Both gene centric and protein centric protein grouping was performed. Quantification was performed as follows: for label-free quantification, protein MS1 precursor area was calculated as the average of the top-three most intense peptides, and peptide MS1 precursor area as the highest PSM area for each peptide. Inferred gene identity false discovery rates were calculated using the picked-FDR method (*Savitski et al., 2015*), whereas the FDR for protein level identities was calculated using the -log10 of best-peptide q-value as a score. The search settings included enzymatic cleavage of proteins to peptides using trypsin limited to fully tryptic peptides. Carbamidomethylation of cysteine was specified as a fixed modification. The minimum peptide length was specified to be six amino acids. Variable modifications (maximum four allowed) were oxidation (of methionine) and phosphorylation (of serine-threonine-tyrosine).

### Proteogenomics search pipeline/SpectrumAI

The MS raw data was searched in customized peptide database including non-synonymous SNPs annotated in CanProVar 2.0 database, the detected variants were curated using the SpectrumAI

pipeline (*Zhu et al., 2018*). In total across the four individuals, we detected 384 unique peptides with SAAV from 215 different proteins where the corresponding SNP was also included on the SNParray. Out of these, 240 SAAV peptides (63%) had genomic support from SNP array. We then removed 11 peptides with N > D, D < N or Q > E, E < Q substitutions since the low mass difference between these amino acids increases the risk of co-isolation of isotopic variants in the MS analysis and generation of false positives.

The genotypes of these detected SNP peptides were determined by SNP array data, and classified into three categories: homo_snp, hetero_snp and wild_type based on the following rules. Wild_type is defined that both alleles are the same to the nucleotide in human reference genome hg19; hetero_snp with one of the allele changed; and homo_snp with both alleles changed. The reference allele sequence for each SNP was extracted from the file *snp142CodingDbSnp.txt* downloaded from UCSC table browser. Variant peptide transfer between mom and baby is inferred when the SNP peptide were detected both in mom and baby, but SNP array genotype data indicates one of them is wild_type and the alternative allele is carried in the other.

## DNA preparation
DNA were prepared from white blood cells using Qiagen DNAeasy blood and tissue kit (product number 69504) according to manufacturer's instructions.

## SNP analysis
Purified DNA was sent to the SNP and SEQ Technology Platform (Uppsala University, Dept. of Medical Sciences, Molecular Medicine, BMC, Husargat. 3, 752 37 Uppsala, Sweden) for genotyping. The analysis was performed using the Illumina Infinium OmniExpressExome-8 v1.4, with 960 919 SNP markers and the Illumina Infinium assay. The results were analyzed using the software GenomeStudio 2011.1 from Illumina Inc. BeadChip type: InfiniumOmniExpressExome-8v1-4_A1 Manifest file: InfiniumOmniExpressExome-8v1-4_A1.bpm, Genome build version: 37 Cluster file ICF InfiniumOmniExpressExome-8v1-4_A1_ClusterFile.egt'.

# Statistical analyses
No statistical method was used to predetermine sample size. Descriptive statistics, t-tests and correlations were performed using GraphPad Prism version 6 (GraphPad Software, La Jolla, CA), Microsoft Excel 2011, and R statistical computing software (https://www.r-project.org). Significance assessed by Student's *t*-test was FDR-adjusted (Benjamini-Hochberg method) to correct for multiple testing. Heatmap visualizations and hierarchical clustering (Pearson correlation) were performed using the online tool Morpheus (software.broadinstitute.org/Morpheus). Annotations were extracted from BioMart (www.ensembl.org/biomart) and Ingenuity Pathway Analysis (IPA) software (QIAGEN Redwood City, www.qiagen.com/ingenuity). Gene Ontology annotation of ranked lists was performed using the online tool GOrilla (Gene Ontology enRIchment anaLysis and visuaLizAtion tool, http://cbl-gorilla.cs.technion.ac.il) (*Eden et al., 2009*; *Eden et al., 2007*). Comparative gene ontology (GO) enrichment analyses of multiple lists were performed using the online tool ToppCluster, https://toppcluster.cchmc.org/publications.jsp (*Kaimal et al., 2010*).

# Protein abundance plots
We examined the presence of tissue leakage proteins, FDA-approved drug targets and cancer-related gene products in plasma by comparing our gene product data with the Human Protein Atlas (www.proteinatlas.org) tissue enriched (enriched defined as at least five-fold higher mRNA levels in a particular tissue as compared to all other tissues, which is a stricter definition than 'tissue enhanced') proteome database. *Figure 3c* has been adapted from: https://commons.wikimedia.org/wiki/File:Female_shadow_anatomy_without_labels.png. In addition, we also mined our data for proteins used in a recently described multi-analyte blood test (CancerSEEK test [*Cohen et al., 2018*]).

# Data availability
Data are available via ProteomeXchange (http://www.proteomexchange.org/) with identifier PXD010899.

## Additional information

### Funding

| Funder | Author |
|---|---|
| Vetenskapsrådet | Maria Pernemalm<br>AnnSofi Sandberg<br>Yafeng Zhu<br>Jorrit Boekel<br>Claes-Göran Östenson<br>Janne Lehtiö |
| Cancerfonden | Maria Pernemalm<br>Janne Lehtiö |
| Stiftelsen för Strategisk Forskning | Maria Pernemalm<br>AnnSofi Sandberg<br>Yafeng Zhu<br>Janne Lehtiö |
| Horizon 2020 Framework Programme | Maria Pernemalm<br>Janne Lehtiö |
| Familjen Erling-Perssons Stiftelse | Janne Lehtiö |
| Barncancerfonden | Janne Lehtiö |
| Stockholms Läns Landsting | Claes-Göran Östenson |
| The Swedish Council for Working Life and Social Research | Claes-Göran Östenson |
| Swedish Diabetes Foundation | Claes-Göran Östenson |
| Stiftelsen Olle Engkvist Byggmästare | Claes-Göran Östenson |

The funders had no role in study design, data collection and interpretation, or the decision to submit the work for publication.

### Author contributions

Maria Pernemalm, Conceptualization, Data curation, Formal analysis, Supervision, Validation, Investigation, Visualization, Methodology, Writing—original draft, Writing—review and editing, Designed the study and the experimental set-up, Performed the proteomics and proteogenomics laboratory work, Performed the data analysis; AnnSofi Sandberg, Conceptualization, Data curation, Formal analysis, Supervision, Validation, Investigation, Visualization, Methodology, Writing—original draft, Writing—review and editing, Designed the study and the experimental set-up, Performed the data analysis; Yafeng Zhu, Conceptualization, Data curation, Formal analysis, Investigation, Visualization, Methodology, Writing—original draft, Writing—review and editing, Performed the mass spectrometry data searches and applied the proteogenomics search pipeline; Jorrit Boekel, Data curation, Formal analysis, Writing—review and editing, Performed the mass spectrometry data searches and applied the proteogenomics search pipeline; Davide Tamburro, Data curation, Formal analysis, Validation, Methodology, Writing—review and editing, Performed the MRM analysis; Jochen M Schwenk, Conceptualization, Formal analysis, Validation, Methodology, Writing—original draft, Writing—review and editing; Albin Björk, Conceptualization, Resources, Writing—original draft, Writing—review and editing, Responsible for the female longitudinal cohort; Marie Wahren-Herlenius, Resources, Writing—review and editing, Responsible for the female longitudinal cohort; Hanna Åmark, Resources, Formal analysis, Methodology, Writing—review and editing, Responsible for the mother and child clinical cohort; Claes-Göran Östenson, Conceptualization, Formal analysis, Methodology, Writing—review and editing, Responsible for the healthy individual cohort; Magnus Westgren, Conceptualization, Supervision, Methodology, Writing—review and editing, Responsible for the mother and child clinical cohort; Janne Lehtiö, Conceptualization, Resources, Supervision, Funding acquisition, Writing—original draft, Writing—review and editing, Designed the study and the experimental set-up

**Author ORCIDs**

Maria Pernemalm ![ORCID] http://orcid.org/0000-0003-4624-031X

Jochen M Schwenk ![ORCID] http://orcid.org/0000-0001-8141-8449

Janne Lehtiö ![ORCID] http://orcid.org/0000-0002-8100-9562

**Ethics**

Human subjects: The plasma collections were approved by local ethics boards and all participants signed informed consent. The approval identifiers for the corresponding studies are as follows; Healthy normals Dnr 91:164 for men and Dnr 95:298 for women, mother/child 2014/1622-31/2 and female longitudinal 2008/915-31/4.

**Decision letter and Author response**

Decision letter https://doi.org/10.7554/eLife.41608.040

Author response https://doi.org/10.7554/eLife.41608.041

## Additional files

**Supplementary files**

• Supplementary file 1. Summary of the number of identifications from the optimization experiment. 1% FDR cutoff at PSM, peptide and protein level, MSGF +with Percolator processing. Label-free, protein centric analysis. * Spiked amount was 4 ng/mL prior to depletion. NA indicates that PSA was not spiked into the sample. ** Percentage of proteins identified by a single peptide, *** Percentage of PSMs originating from the highest abundant protein.

DOI: https://doi.org/10.7554/eLife.41608.019

• Supplementary file 2. Summary of the number of identifications from the optimization experiments, database search allowing for phospho-modifications. Search engines used were Andromeda (Max-Quant) (FDR 1%, PSM and protein level) and MSGF +and Percolator (PSM, peptide and protein level FDR 1%), Protein centric.

DOI: https://doi.org/10.7554/eLife.41608.020

• Supplementary file 3. Summary of phosphodata from the different HiRIEF strip ranges in absolute number and percent. Phosphosite probability calculations were performed within the MaxQuant software. GeLC = gel based (SDS PAGE) separation and digestion.

DOI: https://doi.org/10.7554/eLife.41608.021

• Supplementary file 4. Summary of the overlap analysis between the 3053 proteins identified in the HiRIEF optimization experiments (protein, peptide and PSM data) and the proteins reported in the 2017 version of the PeptideAtlas database. In total 611 genes were uniquely detected by plasma HiRIEF compared to the PeptideAtlas 2017 build and proteins currently targeted with affinity prote-omics assays. Comparative gene ontology enrichment analysis of the HiRIEF unique proteins showed that there was an enrichment of Golgi membrane proteins and MHC complex proteins (provided as an excel file).

DOI: https://doi.org/10.7554/eLife.41608.022

• Supplementary file 5. Gene ontology (GO) enrichment analysis of plasma proteins with low inter-individual variability. The proteins were ranked from smallest to largest inter-individual variation based on their coefficient of variation. The top of the list (least varying) was enriched for proteins linked to humoral activation proteins and receptor activity.

DOI: https://doi.org/10.7554/eLife.41608.023

• Supplementary file 6. Gene ontology (GO) enrichment analysis of plasma proteins with high inter-individual variability. The proteins were ranked from highest to smallest inter-individual variation based on their coefficient of variation. The top of the list (most varying) was enriched for lipid trans-port proteins.

DOI: https://doi.org/10.7554/eLife.41608.024

• Supplementary file 7. Summary of the detection of proteins from the CancerSEEK panel (provided as an excel file).
DOI: https://doi.org/10.7554/eLife.41608.025

• Supplementary file 8. Summary of protein classes used for the analyses in *Figures 3b–c* and *4a–d*. Plasma from the same individual 'not pregnant': six entries, plasma 'third trimester': 30 entries (four entries overlapping between not pregnant and third trimester), in total HPA db placenta enriched: 83 entries).
DOI: https://doi.org/10.7554/eLife.41608.026

• Supplementary file 9. Proteogenomics approach combining SNP analysis and plasma HiRIEF to detect proteins containing SAAV in plasma from two mother-child pairs. Table showing 229 unique variant peptides derived from 161 proteins with DNA support from SNP analysis (provided as an excel file).
DOI: https://doi.org/10.7554/eLife.41608.027

• Supplementary file 10. Brief summary of transfer protein properties in terms of hydrophilicity/hydrophobicity (as evaluated by Gravy score), protein size or pI.
DOI: https://doi.org/10.7554/eLife.41608.028

• Supplementary file 11. Summary of the proteins transferred across the placenta (provided as an excel file).
DOI: https://doi.org/10.7554/eLife.41608.029

• Supplementary file 12. Pooling strategy for the condensed HiRIEF LC-MS/MS analysis used in the longitudinal female cohort.
DOI: https://doi.org/10.7554/eLife.41608.030

• Supplementary file 13. Mirrorplots.
DOI: https://doi.org/10.7554/eLife.41608.031

• Transparent reporting form
DOI: https://doi.org/10.7554/eLife.41608.032

## Data availability

MS raw data are available via ProteomeXchange with identifier PXD010899.

The following dataset was generated:

| Author(s) | Year | Dataset title | Dataset URL | Database and Identifier |
|---|---|---|---|---|
| Pernemalm M, Sandberg A, Zhu Y, Boekel J, Tamburro D, Schwenk JM, Bjork A, Wahren-Herlenius M, Amark H, Ostenson C.G, Westgren M, Lehtiö J | 2019 | Plasma HiRIEF | http://proteomecentral. proteomexchange.org/ cgi/GetDataset?ID= PXD010899 | ProteomeXchange, PXD010899 |

The following previously published datasets were used:

| Author(s) | Year | Dataset title | Dataset URL | Database and Identifier |
|---|---|---|---|---|
| Schwenk JM, Omenn GS, Sun Z, Campbell DS, Baker MS, Overall CM, Aebersold R, Moritz RL, Deutsch EW | 2017 | Human plasma proteome draft | https://db.systemsbiol-ogy.net/sbeams/cgi/Pep-tideAtlas/buildDetails?at-las_build_id=465 | Peptide Atlas, 465 |
| Hasmik Keshishian, Michael W. Burgess, Michael A. Gillette, Philipp Mertins, Karl R. Clauser, D. R. Mani, Eric W. | 2015 | Sensitive Biomarker Discovery in Plasma Yields Novel Candidates for Early Myocardial Injury | https://massive.ucsd. edu/ProteoSAFe/data-set.jsp?task= 9c3f5d8472c1486a8fce-da556598ac94 | CCMS, MSV0000790 33 |

Kuhn, Laurie A.
Farrell, Robert E.
Gerszten and Steven A. Carr

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
