## [Decision Letter]

Thank you for sending your article entitled "In-depth human plasma proteome analysis captures tissue proteins and transfer of protein variants across the placenta" for peer review at *eLife*. Your article has been evaluated by three peer reviewers, and the evaluation has been overseen by a Reviewing Editor and Harry Dietz as the Senior Editor.

Summary:

The paper provides new avenues to deep human plasma profiling, and applies this multi-dimensional fractionation technology to the analysis of plasma proteins crossing the placenta of pregnant women. The paper provides practical guides to the capabilities of high-resolution isoelectric focusing gel strips to mass spectrometry to drill deep into the plasma proteome to a level of 1888 proteins per sample and 3053 proteins in total. The technical aspects are well discussed and compared to public databases to provide a context of the protein depth. The analysis of both mother and fetus derived proteins by the notion of identifying peptides harboring SAAV's from non-synonymous SNP differences is a key feature of this work to unequivocally identify proteins that will have to have transferred across the placenta and entered into each other's circulation.

Opinion:

This work provides a valuable advancement for mass spectrometry based proteomic analysis of the plasma proteome in terms of sensitivity and depth of coverage. In addition it could be an important resource to the field. However, major issues, described below, should be carefully dealt with prior publication.

Essential revisions:

1) Novelty compared to the state of the art: There are several prior reports on use of the method for plasma analysis dating back to Heller et al., 2005, and the authors own work published in 2014 in Nature Methods provides much of the detail off-line HiRIEF separations. The first step of using abundant protein depletion to gain depth in the plasma proteome is commonly used and common knowledge. One new aspect reported relative to the authors prior published work is that broad range IEF strips outperform narrow-range IEF strips with respect to the numbers of peptides and proteins identified. This result contradicts the author's own prior results obtained with cell and tissue samples. This important difference between the behavior of cells/tissues and plasma should be described and further explored as it represents an interesting and potentially novel finding.

- The section on exploring the proteome demonstrates that the HiRIEF method can replicate already known finding about the differences in the plasma proteome between males and females.

- The section on protein enriched during pregnancy and enriched in cord blood is interesting, but largely describes what is already known from other blood analysis papers. Perhaps most interesting, although again simply descriptive, is the finding of fetal hemoglobin in maternal blood of some of the subjects. No mechanistic or biological implications for this are presented.

- Variant observation – it is well known that MS-based proteogenomics can do this (see authors own work in 2014; and, for example, Bing Zhang Nature 2014; Mertins Nature 2016; Hui Zhang Cell 2016). The fact that one can do this in blood and show correlation to the genome is therefore fully expected.

2) Method analyses 72 fractions from the HiRIEF separation and uses ca. 89 hours of MS instrument time to acquire the data. The number of proteins reported is ca. 1/3 of what has been achieved with slightly greater instrument use time (Branca et al., 2014 and Keshishian et al., 2016). The HiRIEF method does yield some proteins not detected by other approaches, but overall the depth is limited in comparison to the Keshishian method which makes it somewhat less relevant for biomarker discovery or to inform biology.

3) Importantly, the method appears to not be particularly reproducible as many of the supposedly confidently identified proteins must only be being observed in one or a small number of analyses: "When applying the HiRIEF methodology we identified on average 1505 proteins (per sample)…."; "In total 3053 proteins (across at least 16 experiments, perhaps more…hard to tell) were identified in the different experiments.' The fact that this aggregate number approaches the 3509 proteins from the most recent version of the human plasma atlas is, therefore, not of much use from the perspective of what the method can do on an individual sample basis for biology or biomarker discovery. What matters is what can be confidently and repeatedly detected across multiple patient/subject samples.

4) Importantly, the number of proteins confidently identified by the authors appears to be overstated. From 1/4 to 1/2 of the proteins claimed to be identified are identified on the basis of a single peptide according to Supplementary file 1. It is standard practice in proteomics to only claim confident identification of a protein when 2 or more unique peptides from that protein are identified. Therefore, the large number of proteins claimed (ca. 3000 across multiple experiments) is likely highly inflated as it appears to include "one-hit wonders". It also appears that these numbers derive from analysis of the same plasma pool rather than plasma from different subjects – this needs to be clearly stated in the text.

5) The PSA spike experiment is interesting, but simply replicates what is already known: that if you know the fraction that peptide(s) from a protein of interest elute in, you can get significantly higher sensitivity by focusing experiments on those selected fractions. In this case the authors changed to selected fractions a single narrow pI range. This is similar in concept to the PRISM method developed at PNNL by the Smith group.

6) Authors state that the "method" is reproducible. As noted above, based on the ability to confidently and repeatedly detect proteins appears to be rather low. Exactly which parts are reproducible (i.e., depletion, IEF fractionation, LC-MS/MS, identifications, etc.) is not entirely clear. The data used to support this statement are TMT intensity ratios obtained in a set of 4 x TMT 10-plex experiments, and focuses on a subset of ca. 1000 proteins found to be in common across the experiments. What is not demonstrated well (or at all) is the ability to confidently determine differences in protein abundance between samples. It would be much more convincing to show the reproducibility of the relative abundance differences of proteins from different patient samples and which cover the detection range from high to low abundance. Data plotted appear to be just for the pooled plasma sample in each of the 4 TMT plexes where no differences in abundance are expected across samples for the same protein. Perhaps more importantly, the actual method used for analysis of subject plasma samples does not use TMT labeling, so the reproducibility of the label-free proteomics approach has not being directly evaluated.

7) Keshishian et al. report a TMT1- method that provides ca. 600 proteins/sample in ca. 5h of analysis time (Keshishian et al., 2016). This paper should be cited and your results compared.

8) Please provide tables: of proteins which are purported to be newly discovered in plasma (n=611); protein affinity assay to provide the identity of the 751 protein assays that are not in the HiRIEF or PeptideAtlas lists, and of whom they are available from; CancerSEEK fingerprint used so readers do not have to go to the original paper to find out this information.

9) The figures are a little on the small size of acceptable quality and care should be taken to provide legible axis and text within the figure. If they can be made larger, it would benefit the paper immensely.

10) For the identification of new proteins, full identification data provided as tables should provide the MS spectral details of each of the new plasma proteins identified, the MS spectra of peptides with SAAV's, and provide the genomic data used to construct the SAAV databases for proteomics search algorithms.

11) Choosing a single amino acid variation (SAAV) as the strategy to distinguish between mother and fetus proteins might hold a weakness. Since the mass difference between the WT peptide and the variant peptide is relatively low (based on one amino acid substitution) it may be explained by additional post-translational modifications (PTMs) from ones that were already mentioned and may not necessarily be explained by SAAV. For example, in Supplementary file 9: peptide 2 (T>S) – the mass difference equals Methylation, peptides 7,25,26,30,32 (V>I/L) – mass difference equal Methylation, peptide 13,15 (E>D) – mass difference equal Methylation, peptide 1,18,19 (P>L) – mass difference almost equals oxidation. We would suggest to closely review the peptide spectra to validate that the source of the different mother-fetus peptides is due to genomic variance and not PTM. This should be addressed or discussed in the manuscript and it may limit the interpretation of transfer.

12) When analyzing the inter-individual variability of the plasma proteome (Figure 2B) the authors identified HLA proteins in their top 20 list. This may be due to the highly polymorphic nature of HLA causing individual's peptides to not be properly assigned to the reference sequence. This is worth considering or mentioning.

13) The authors should clarify what they mean by 'load'. Is this the amount of protein loaded onto the MARS14 column or onto the isoelectric focusing strips. If the latter, did it require multiple depletions of the same sample to reach the amount of total protein?

14) In the Methods section they list two types of data processing tools (MS-GF+ and MaxQuant). It is not clear why did they use two different tools and for which samples did they use each tool?

[Editors' note: further revisions were requested prior to acceptance, as described below.]

Thank you for resubmitting your work entitled "In-depth human plasma proteome analysis captures tissue proteins and transfer of protein variants across the placenta" for further consideration at *eLife*. Your revised article has been favorably evaluated by Harry Dietz as the Senior Editor, a Reviewing Editor, and two reviewers.

The manuscript has been substantially improved but there are some remaining issues that need to be addressed before final acceptance, as outlined below.

1) Figure 3B – it is unclear if the colored dots obscure one another – is there a way to statistically strengthen this statement? Perhaps it would be useful to calculate the average abundance of each of the four groups and compare them? It seems that the statement is not supported statistically.

2) Overall, the authors have improved the manuscript substantially and the paper is almost ready for acceptance with the new information provided. However, the revised manuscript could have been better with tables that have the required information contained in them as requested, as well as the individual peptide spectral files from all of the novel identifications purported and provided to be more extensive and user-friendly than the current revised tables included. If done, many readers can take the information provided in this paper and perform their own analysis and corroborate the findings described within. For example, the review asked for tables of identification and spectral information of each novel peptide to describe the new protein lists in both the plasma identifications as well as the variant peptides that were found with this work, yet a simplistic table of ensemble identifiers (Supplementary file 4) was only provided for the novel plasma proteins even though the full spectral information was provided for the variant peptides. Given the scrutiny that the work presented in this manuscript will come under from the research community interested in plasma protein analysis, it is beholden to the authors to defend the work to the best of their capabilities as it stands if the work is to be published in a quality journal. The tables of new identifications would encompass all the technical details of the proteins discovered including web-linked accession number to not only ensemble but also to neXtProt and PeptideAtlas given this was one of the databases referred to, how many peptides for each novel protein were found, their individual score and probabilities, as well as any other quality factors used for their positive identification. The PSM MS spectrum plots were not provided as requested for the 611 newly discovered plasma proteins, however, these were supplied for the mother/baby variant peptides identified in the demonstration project of this paper. To complete the resubmission, please provide this updated table (Supplementary file 4) also identifying which files in the raw data repository of this work the novel identifications were made from so readers can quickly get at the data if needed.

---

## [Author Response]

Essential revisions:1) Novelty compared to the state of the art: There are several prior reports on use of the method for plasma analysis dating back to Heller et al., 2005, and the authors own work published in 2014 in Nature Methods provides much of the detail off-line HiRIEF separations. The first step of using abundant protein depletion to gain depth in the plasma proteome is commonly used and common knowledge. One new aspect reported relative to the authors prior published work is that broad range IEF strips outperform narrow-range IEF strips with respect to the numbers of peptides and proteins identified. This result contradicts the author's own prior results obtained with cell and tissue samples. This important difference between the behavior of cells/tissues and plasma should be described and further explored as it represents an interesting and potentially novel finding.

The aim of the current study was to develop and apply a robust in-depth plasma proteomics workflow, capable of analyzing clinically relevant sized cohorts, while still retaining the analytical depth to reach low abundant tissue leakage proteins and potential biomarkers.

We agree with the reviewers that the use of isoelectric focusing (IEF) for separation prior to mass spectrometry (MS) -based proteomics is not a novelty itself. However, adapting the high-resolution peptide IEF for plasma has not been demonstrated previously. The previously published work from Heller et al. has applied OFF-gel fractionation on a protein and peptide level as well as in combination with high abundant protein depletion to identify 81 proteins. Here, we elucidate which pI ranges that are most beneficial for the special characteristics of the plasma proteome in terms of proteome coverage and analytical depth.

We describe a workflow with capability of reproducible analysis of multiple plasma samples by use of HiRIEF pI-based plasma fractionation on ultra-narrow, narrow and wide ranges. We show that the benefit of wide range lies in the increased number of PSMs and peptides, which improves quantification accuracy

– a key factor importance to biomarker discovery with inherent high variation between samples. The benefit of ultra-narrow lies in the increased number of protein identifications in the low protein abundance range (Figure 1—figure supplement 1 and the “PSA example”). The pH range around 4 is very dense in peptides, and hence protein identification of low abundant proteins can benefit from the additional fractionation by the ultra-narrow strips in that region as shown in the PSA spike-in experiment.

We report over 600 novel proteins previously not incorporated in Peptide Atlas plasma proteomics database adding significant number of proteins, and peptides for targeted proteomics, to the public resource. We also show that the identifications are robust enough to cover single amino acid variants in the plasma proteome, which has previously not been shown.

Further, we show that the plasma HiRIEF method can detect tissue leakage proteins in untargeted proteomics (as exemplified by the placental proteins specifically identified during pregnancy). We also show, for the first time, that we can use plasma proteogenomics to follow peptides containing single amino acid variants across the placenta during pregnancy.

In the new version of the manuscript a new section in the Introduction has been added highlighting the difference between the plasma and tissue/cellular proteome. Data from cell lines separated by HiRIEF has been added to Table 1 for comparison of the cellular analysis in Zhu et al., 2018.

- The section on exploring the proteome demonstrates that the HiRIEF method can replicate already known finding about the differences in the plasma proteome between males and females.

To benchmark the plasma HiRIEF method performance against current knowledge on the plasma proteome we have chosen to highlight findings that is supported by other studies on plasma. The aim with the benchmarking is to show the validity of the results from the method. The previously reported differences in the plasma proteome between men and women is used as an important reference analysis to demonstrate that the method can replicate previous observations shown by other techniques.

- The section on protein enriched during pregnancy and enriched in cord blood is interesting, but largely describes what is already known from other blood analysis papers. Perhaps most interesting, although again simply descriptive, is the finding of fetal hemoglobin in maternal blood of some of the subjects. No mechanistic or biological implications for this are presented.

This is a good point that is missing in the current version of the manuscript. The expression of fetal hemoglobin among adults has been reported previously and can partly be explained by a hereditary component (see e.g. Thein et al., 2007 and Galarneau et al., 2010). A short section describing this and the references has been added to this section (subsection “Tissue annotated proteins detected in plasma”, last two paragraphs). However, we see the novelty here in detecting proteins transferring cross placenta unequivocally identified.

- Variant observation – it is well known that MS-based proteogenomics can do this (see authors own work in 2014; and, for example, Bing Zhang Nature 2014; Mertins Nature 2016; Hui Zhang Cell 2016). The fact that one can do this in blood and show correlation to the genome is therefore fully expected.

While it is true that both we and others have shown variant detection by MS-based proteomics in cells and tissues, we also concluded in Branca et al., 2014, that a prerequisite in variant detection is in-depth coverage of proteome and peptidome. Given the analytical difficulties associated with plasma proteome analysis it cannot necessarily been taken for granted that variants expressed by different organs of the human body can be detected, as it for other sample types such as cell cultures, particularly considering the skewed distribution of proteins concentrations.

Here we have applied the detection of variants to study proteins passing across the placenta. This opens up a new possibility to study fundamental questions linked to fetal-maternal communication in vivo.

2) Method analyses 72 fractions from the HiRIEF separation and uses ca. 89 hours of MS instrument time to acquire the data. The number of proteins reported is ca. 1/3 of what has been achieved with slightly greater instrument use time (Branca et al., 2014 and Keshishian Nature Protocols 2016). The HiRIEF method does yield some proteins not detected by other approaches, but overall the depth is limited in comparison to the Keshishian method which makes it somewhat less relevant for biomarker discovery or to inform biology.

In the first version of the manuscript, a comparison with the Keshishian dataset from 2015 (Branca et al., 2014) was only made indirectly, as it is one of the datasets included in the latest version of the Plasma Proteome database compiled within the PeptideAtlas (Figure 1—figure supplement 2, Schwenk et al., 2017).

We fully agree that work by Keshishian et al. is an important benchmark in the field of plasma proteomics and have therefore added a head to head comparison with their method and the data from the MCP paper from 2015. The raw data from the Nature protocols paper by Keshishian et al. is not available deposited in the repository, so no direct comparative analysis with the novel data sets from that publication has been performed. We acknowledge that there is a growing diversity of MS-based approaches and that these require to highlight their added value in relation to the benchmarks of the field.

In the initial response to the editor we included several preliminary analyses where we had re-searched the available raw data from Keshishian et al. using same search parameters as for our HiRIEF analysis. After feedback from the editor, these preliminary analyses were not added in updated version of the manuscript. Instead, we have introduced a short section using the re-searched data to show a core set of proteins that can be confidently identified across the different experiments described in Keshishian et al., 2015 and compared it to the core set generated by the HiRIEF method, as we believe this is a key feature in the analysis of larger clinical cohorts and a core set analysis was not reported in the original MCP paper (subsection “Performance assessment of the plasma HiRIEF method”).

In addition, we have also added a head to head comparison of the main results describing the performance of the method from Keshishian et al. as reported by the authors from the MCP study in terms of sample load, CV, MS time and fractionation (see the aforementioned subsection).

In the revised version of the manuscript we have added a second TMT-cohort which is analyzed in a condensed HiRIEF fraction pooling approach to reduce the MS time to approx. 55h. We show that the reduction in MS-time does not affect the number of identified proteins, showing improved throughput of the HiRIEF method. In addition, the amount of plasma per sample required in the HiRIEF TMT-10 experiment is 40uL vs approx. 400uL used for the in-depth analysis in the Keshishian et al.papers.

A key feature in the comparison between the method described by Keshishian et al. and the HiRIEF method is the applicability to clinical cohorts including samples from a number of different individuals. We acknowledge that Keshishian et al. report higher number of identified proteins, but we argue that the HiRIEF method has an advantage for in-depth analysis of larger patient cohorts – using less material, less MS analysis time and showing robust identifications across two different cohorts with 42 different individuals, sampled at different times and different hospitals.

3) Importantly, the method appears to not be particularly reproducible as many of the supposedly confidently identified proteins must only be being observed in one or a small number of analyses: "When applying the HiRIEF methodology we identified on average 1505 proteins (per sample)…."; "In total 3053 proteins (across at least 16 experiments, perhaps more…hard to tell) were identified in the different experiments.' The fact that this aggregate number approaches the 3509 proteins from the most recent version of the human plasma atlas is, therefore, not of much use from the perspective of what the method can do on an individual sample basis for biology or biomarker discovery. What matters is what can be confidently and repeatedly detected across multiple patient/subject samples.

While setting up and optimizing the workflow, an average 1505 proteins were identified per experimental condition using different loads and different pI ranges. The pI ranges cover different pI areas and are in that sense complementary and hence are not expected to identify exactly the same peptides and proteins (see Figure 1—figure supplement 1B for overlap between pI ranges). This flexibility is indeed an advantage of the method (compared to e.g. high-pH reversed phase which is less flexible in terms of orthogonality). In the new version of the manuscript we have added a table (Table 1) summarizing the findings from the optimization and updated the text to clarify the benefit of being able to use different pH intervals for different analytical purposes.

In addition, we have added a second cohort to make a core-set analysis of proteins that can be confidently identified across samples, TMT-sets and cohorts. In this gene-centric core set analysis we show that 828 genes can be confidently identified across nine TMT-sets (subsection “Performance assessment of the plasma HiRIEF method”).

4) Importantly, the number of proteins confidently identified by the authors appears to be overstated. From 1/4 to 1/2 of the proteins claimed to be identified are identified on the basis of a single peptide according to Supplementary file 1. It is standard practice in proteomics to only claim confident identification of a protein when 2 or more unique peptides from that protein are identified. Therefore, the large number of proteins claimed (ca. 3000 across multiple experiments) is likely highly inflated as it appears to include "one-hit wonders". It also appears that these numbers derive from analysis of the same plasma pool rather than plasma from different subjects – this needs to be clearly stated in the text.

For protein identifications we have used MSGF+ and Percolator within our standard pipeline. In the current manuscript we have been using a strict 1% FDR cut off at PSM, peptide and protein level. In addition, the open source commonly used MaxQuant search engine Andromeda was used for quality control comparison and for calculating modification probabilities and location.

We do not agree with the reviewer that one-peptide identifications are inherently poor-quality identifications. Rather, for peptide identification, currently the most accepted result validation method of the quality of the peptide identification is through false discovery rate (FDR) of each individual peptide. This has been well described in the paper by Pevzner et al. (False Discovery Rates of Protein Identifications: A Strike against the Two-Peptide Rule, Pevzner et al., JPR 2008).

Also see reply to point 3 above regarding number of identified proteins in different experiments and the core set analysis.

5) The PSA spike experiment is interesting, but simply replicates what is already known: that if you know the fraction that peptide(s) from a protein of interest elute in, you can get significantly higher sensitivity by focusing experiments on those selected fractions. In this case the authors changed to selected fractions a single narrow pI range. This is similar in concept to the PRISM method developed at PNNL by the Smith group.

While PRISM method is indeed splendid, however, there are key differences between the methods. The advantage with the HiRIEF methodology for targeted MS is that it uses inherent peptide property that is peptide sequence dependent, the pI of a peptide, which can be predicted. Hence, for any tryptic peptide with known sequence, we can use the pI predictor to find the fraction in which the peptide will be found if present above detection limits. The fractionation method can therefore be optimized a priori based on the calculated peptide pI and do not rely on indexing of large previously analyzed experimental datasets. Moreover, the prediction accuracy of where a peptide ends up in LC fractionation is lower (property used in PRISM), especially on variant level detection, due to that it suffers on variation due to inter- chromatographic interactions depending on sample compositions (Moruz et al. MassSpectrometry reviews, 2016).

6) Authors state that the "method" is reproducible. As noted above, based on the ability to confidently and repeatedly detect proteins appears to be rather low. Exactly which parts are reproducible (i.e., depletion, IEF fractionation, LC-MS/MS, identifications, etc.) is not entirely clear. The data used to support this statement are TMT intensity ratios obtained in a set of 4 x TMT 10-plex experiments, and focuses on a subset of ca. 1000 proteins found to be in common across the experiments. What is not demonstrated well (or at all) is the ability to confidently determine differences in protein abundance between samples. It would be much more convincing to show the reproducibility of the relative abundance differences of proteins from different patient samples and which cover the detection range from high to low abundance. Data plotted appear to be just for the pooled plasma sample in each of the 4 TMT plexes where no differences in abundance are expected across samples for the same protein. Perhaps more importantly, the actual method used for analysis of subject plasma samples does not use TMT labeling, so the reproducibility of the label-free proteomics approach has not being directly evaluated.

This is a valid point and addressing it will help us to improve the manuscript. As we see this, the reproducibility can be divided into two categories: quantitative reproducibility and reliability of repeated detection of overlapping proteome between analyses. The pooled internal standards that have been used for the quantitative reproducibility calculations have been pooled after the digestion of the proteins, meaning that only technical variability post digestion is taken in to account.

In the current version we have clarified which steps are included in the reproducibility calculations (subsection “Performance assessment of the plasma HiRIEF method). We have also added a core set analysis of proteins that can be repetitively detected across different TMT sets and cohorts using the 3-10 strip.

For quantitative analysis using TMT-labelling we have used that in the analysis looking a proteins with high and low variability in the plasma proteome and differences between abundance between men and women.

7) Keshishian et al. report a TMT1- method that provides ca. 600 proteins/sample in ca. 5h of analysis time (Nature Protocols 2016). This paper should be cited and your results compared.

We agree, see reply under point 2.

8) Please provide tables: of proteins which are purported to be newly discovered in plasma (n=611); protein affinity assay to provide the identity of the 751 protein assays that are not in the HiRIEF or PeptideAtlas lists, and of whom they are available from; CancerSEEK fingerprint used so readers do not have to go to the original paper to find out this information.

We have added a supplementary table (Supplementary file 4) describing the proteins from the overlap analysis as well as a supplementary file describing the identified proteins from on the cancer seek panel (Supplementary file 7). The protein assays are described in the original publication by Schwenk et al.

9) The figures are a little on the small size of acceptable quality and care should be taken to provide legible axis and text within the figure. If they can be made larger, it would benefit the paper immensely.

We agree with the reviewer and have enlarged and simplified Figure 1 and split Figure 3 into two separate figures, as well as enlarged several of the supplementary figures.

0) For the identification of new proteins, full identification data provided as tables should provide the MS spectral details of each of the new plasma proteins identified, the MS spectra of peptides with SAAV's, and provide the genomic data used to construct the SAAV databases for proteomics search algorithms.

The construction of the VarDB database is described in detail in Zhu et al., 2018. The reference and information has been added in the updated version.

Mirror plots, showing spectra with DNA support (from ‘donor’) and without DNA support (from ‘recipient’), for the transferred peptides are included in a supplementary file (Supplementary file 13). We have manually inspected the mirror plots and highlighted those where we believe there is a poor match (n=5) and updated the main text accordingly (subsection “Plasma proteogenomics”).

11) Choosing a single amino acid variation (SAAV) as the strategy to distinguish between mother and fetus proteins might hold a weakness. Since the mass difference between the WT peptide and the variant peptide is relatively low (based on one amino acid substitution) it may be explained by additional post-translational modifications (PTMs) from ones that were already mentioned and may not necessarily be explained by SAAV. For example, in Supplementary file 9: peptide 2 (T>S) – the mass difference equals Methylation, peptides 7,25,26,30,32 (V>I/L) – mass difference equal Methylation, peptide 13,15 (E>D) – mass difference equal Methylation, peptide 1,18,19 (P>L) – mass difference almost equals oxidation. We would suggest to closely review the peptide spectra to validate that the source of the different mother-fetus peptides is due to genomic variance and not PTM. This should be addressed or discussed in the manuscript and it may limit the interpretation of transfer.

To curate the variant data, we have used the iPAW pipeline described in Zhu et al., 2018. Of particular importance in IPAW is SpectrumAI (Automated Inspection), a tool that curates single amino acid substituted peptides by requiring ions to directly support the residue substitutions in MS/MS spectra. SpectrumAI will make sure that the mass change is exactly on that amino acid position and not elsewhere.

We agree that if the residue mass in a position of D>E substitution is a genuine glutamate or it is an aspartate that got a post-translational modification of a methylation in the cell, we can never be 100% sure. However, T>S (peptide 2) would be loss of methyl group, not gain, and that modification is not known at all (and is absent from the unimod database).

Moreover, methylations are unlikely in S, V, D, or P. There are papers describing methylation on R and K, for example, but not on S, V, D or P. To our knowledge only methyl-D (PMID: 18220335) has been described, and even in that paper out of a total of 7000 peptides identified in MS, they found 3 methyl-D peptides.

Although these alternative explanations for the spectra could be possible, we believe that they less likely than a SAAV with genomic support.

To supplement the sequence confirmation of the transferred variants we have included mirror plots showing the peptide spectrum from the donor (with sequence support) mirrored against the postulated same transferred peptide in the recipient (without sequence support), and we have also manually inspected the spectra and highlighted those of poor quality. See reply to point 10.

12) When analyzing the inter-individual variability of the plasma proteome (Figure 2B) the authors identified HLA proteins in their top 20 list. This may be due to the highly polymorphic nature of HLA causing individual's peptides to not be properly assigned to the reference sequence. This is worth considering or mentioning.

This is a good point and we have added it to the revised version of the manuscript (subsection “Exploring the plasma proteome”).

13) The authors should clarify what they mean by 'load'. Is this the amount of protein loaded onto the MARS14 column or onto the isoelectric focusing strips. If the latter, did it require multiple depletions of the same sample to reach the amount of total protein?

The load is referring to the peptide amount loaded onto the IEF strips. This and other lab jargon terminologies has been clarified in the text (subsection “Optimizing the HiRIEF method for plasma analysis”, as well as in Figure 1).

14) In the Materials and methods section they list two types of data processing tools (MS-GF+ and MaxQuant). It is not clear why did they use two different tools and for which samples did they use each tool?

We used MSGF+ and Percolator within our standard pipeline. The open source commonly used MaxQuant search engine Andromeda was used for quality control comparison and for calculating modification probabilities and location.

[Editors' note: further revisions were requested prior to acceptance, as described below.]

The manuscript has been substantially improved but there are some remaining issues that need to be addressed before final acceptance, as outlined below.1) Figure 3B – it is unclear if the colored dots obscure one another – is there a way to statistically strengthen this statement? Perhaps it would be useful to calculate the average abundance of each of the four groups and compare them? It seems that the statement is not supported statistically.

We would like to thank the reviewers and editors with thorough work on reviewing the manuscript and we are happy to hear that the revisions that has been made to the manuscript is appreciated by the reviewers. We also agree that all dots in Figure 3B are not separated and hence could mask each other. We have hence added a novel figure supplement (Figure 3—figure supplement 1) showing the distributions in more detail and have added statistical analyses comparing the different distributions to each other to clarify the differences between classical plasma proteins and other protein groups referred to in the text.

2) Overall, the authors have improved the manuscript substantially and the paper is almost ready for acceptance with the new information provided. However, the revised manuscript could have been better with tables that have the required information contained in them as requested, as well as the individual peptide spectral files from all of the novel identifications purported and provided to be more extensive and user-friendly than the current revised tables included. If done, many readers can take the information provided in this paper and perform their own analysis and corroborate the findings described within. For example, the review asked for tables of identification and spectral information of each novel peptide to describe the new protein lists in both the plasma identifications as well as the variant peptides that were found with this work, yet a simplistic table of ensemble identifiers (Supplementary file 4) was only provided for the novel plasma proteins even though the full spectral information was provided for the variant peptides. Given the scrutiny that the work presented in this manuscript will come under from the research community interested in plasma protein analysis, it is beholden to the authors to defend the work to the best of their capabilities as it stands if the work is to be published in a quality journal. The tables of new identifications would encompass all the technical details of the proteins discovered including web-linked accession number to not only ensemble but also to neXtProt and PeptideAtlas given this was one of the databases referred to, how many peptides for each novel protein were found, their individual score and probabilities, as well as any other quality factors used for their positive identification. The PSM MS spectrum plots were not provided as requested for the 611 newly discovered plasma proteins, however, these were supplied for the mother/baby variant peptides identified in the demonstration project of this paper. To complete the resubmission, please provide this updated table (Supplementary file 4) also identifying which files in the raw data repository of this work the novel identifications were made from so readers can quickly get at the data if needed.

We agree with the reviewers that the availability of the MS data is of outmost importance and all raw data as well as all result files (protein, peptide and psm tables) from the MS analysis is available in the public repository proteomeXchange as described in the Materials and methods section.

With acceptance of the publication the data will be made available for the scientific community.

In response to review comment 2 we have added three novel tables with more detailed information about the proteins identified in the optimization as supplementary information (updated Supplementary file 4).

Supplementary file 4 is now updated and contains all 3053 proteins that were identified in the optimization, including data on number of peptides, #psms, q-value etc. A column indicating all the 611 proteins that has not been reported in PeptideAtlas has also been added.

In addition, the updated Supplementary file 4 also contains all the peptides that has been identified in the optimization, as well as details on #psms, q-value etc. A column indicating which peptides that are mapping to the 611 proteins that has not been reported in PeptideAtlas is also included.

Last the last novel table within Supplementary file 4 contains all the PSMs mapping to the 611 proteins that has not been reported in PeptideAtlas and information on in which raw file(s) they can be found and the corresponding scan numbers (spectrum).

For the transferred peptides we have also provided annotated spectrum, however we respectfully argue against a figure on the over 40 000 spectrum matching to the 611 novel proteins as a supplementary figure, and even if only selected spectra showing the specific peptides for each protein would be included, the file would be of limited use for the community. In our opinion, having access to the raw data, and the added information on the tables to guide researcher to find the spectra, provides the readers with a much more flexible format and the opportunity for the scientific community to re-analyze and evaluate the data much more thoroughly.

In our reporting of the mass spectrometry data we have followed the Miape guidelines (minimal information about a proteomics experiment) http://www.psidev.info/miape postulated by the Human Proteome Organisation where *reference* to each spectrum is required (included for each search in all psm files on proteomeXchange).